# Population-level comparisons of gene regulatory networks modeled on high-throughput single-cell transcriptomics data

Daniel Osorio [1] ✉, Anna Capasso[1], S. Gail Eckhardt[1], Uma Giri[1], Alexander Somma[1], Todd M. Pitts[2], Christopher H. Lieu[2], Wells A. Messersmith[2], Stacey M. Bagby[2], Harinder Singh[3], Jishnu Das [3], Nidhi Sahni[4,5], S. Stephen Yi [1,6,7,8] ✉ & Marieke L. Kuijjer [9,10,11] ✉

Single-cell technologies enable high-resolution studies of phenotype-defining molecular mechanisms. However, data sparsity and cellular heterogeneity make modeling biological variability across single-cell samples difficult. Here we present SCORPION, a tool that uses a message-passing algorithm to reconstruct comparable gene regulatory networks from single-cell/nuclei RNA-sequencing data that are suitable for population-level comparisons by leveraging the same baseline priors. Using synthetic data, we found that SCORPION outperformed 12 existing gene regulatory network reconstruction techniques. Using supervised experiments, we show that SCORPION can accurately identify differences in regulatory networks between wild-type and transcription factor-perturbed cells. We demonstrate SCORPION's scalability to population-level analyses using a single-cell RNA-sequencing atlas containing 200,436 cells from colorectal cancer and adjacent healthy tissues. The differences between tumor regions detected by SCORPION are consistent across multiple cohorts as well as with our understanding of disease progression, and elucidate phenotypic regulators that may impact patient survival.

In eukaryotes, gene expression is carefully regulated by transcription factors[1], which are proteins that play a crucial role in determining cell identity and controlling cellular states. They achieve this by either activating or repressing the expression of specific target genes. This regulation is dependent on the abundance of transcription factors, their ability to bind to chromatin (DNA–protein complex) and various post-translational modifications they undergo[2]. It is well known that changes in regulatory interactions may result in abnormal expression profiles and diseased phenotypes[3]. Typically, gene regulatory networks are constructed and compared to identify mechanistic alterations in the relationship between transcription factors and their target genes that result in these abnormal phenotypes[4]. Transcriptomic data can be used to infer gene regulatory networks by examining the co-expression patterns of genes that are part

of the same regulatory programs. Depending on the set of cells or samples with transcriptomic data included in the gene regulatory network reconstruction, networks can either represent the regulatory programs of specific cell types within a tissue, or capture average mechanisms that define the entire tissue from which the sample was taken[5].

Using the gene expression variability found in RNA-sequencing (RNA-seq) data from single cells/nuclei, it is possible to infer gene regulatory networks for each cell type or cell state within a single sample[6]. However, when multiple samples are available, transcriptomes from different samples are typically collapsed by an experimental group before the group-level comparison is carried out. In the context of differential network analysis, an aggregate network is often constructed by combining the transcriptomes of all cells

within each experimental group. This network then represents the characteristics of each experimental group, and these aggregate network models can be used for comparative analysis[7]. To learn more about the transcription factor–target gene interactions that support the phenotype of interest, this network is scrutinized or compared with others[8]. Although useful, aggregate network models are not designed to account for evaluating regulatory heterogeneity between samples[9].

Pseudo-bulk profiles are frequently calculated in differential gene expression analysis to take into account biological variation between samples[10]. However, to identify consistent mechanistic patterns causing phenotypic changes across samples within a population, the biological variability between transcription factors and their target gene interactions should ideally be modeled across multiple samples[9]. This entails developing time-efficient techniques for constructing highly accurate and comparable gene regulatory networks from single-cell/nuclei RNA-seq data.

Using high-throughput RNA-seq data from single cells or nuclei to create comparable gene regulatory networks is a difficult task. This type of data is highly sparse and frequently contains information based on multiple cellular states in a single experiment, making sample comparison challenging. Furthermore, non-biological factors frequently affect data during library preparation, reducing our ability to detect biologically accurate correlation structures[5]. For example, the high level of sparsity in single-cell RNA-seq data limits the application of methods originally designed for gene regulatory network construction using bulk RNA-seq data that use correlation across samples to estimate network interactions. This includes methods that solely use correlation metrics over sparse matrices to model regulatory interactions, such as Weighted Correlation Network Analysis (WGCNA), as well as methods that incorporate prior information on gene regulation to estimate regulatory interactions such as PANDA (Passing Attributes between Networks for Data Assimilation)[11].

To address these challenges in differential gene regulatory network analyses on single-cell data, we present SCORPION (Single-Cell Oriented Reconstruction of PANDA Individually Optimized gene regulatory Networks), a tool that uses coarse-graining of single-cell/nuclei RNA-seq data to reduce sparsity[12] and improve the ability to detect correlation structures in these data. The coarse-grained data generated are then used to reconstruct gene regulatory networks, using the regulatory network reconstruction algorithm (PANDA)[11]. PANDA uses a message-passing approach to integrate multiple sources of information, such as protein–protein interaction, gene expression and sequence motif data, to predict regulatory relationships. Owing to the coarse-graining and the use of the same baseline priors for each aggregated Super/MetaCell, SCORPION can reconstruct comparable, fully connected, weighted and directed transcriptome-wide gene regulatory networks suitable for statistical analyses that leverage multiple samples per experimental group—something we refer to in the remainder of this paper as 'population-level studies.'

We tested the performance of SCORPION's coarse-grained input data for network modeling using synthetic data via BEELINE, a tool for systematically evaluating cutting-edge algorithms for inferring gene regulatory networks from single-cell transcriptional data[13]. We found that networks modeled on data desparsified with SCORPION outperformed 12 other gene regulatory network reconstruction techniques across 7 metrics. In addition, using supervised experiments, we show that SCORPION can precisely identify biological differences in regulatory networks between wild-type cells and cells carrying transcription factor perturbations. Furthermore, we demonstrate SCORPION's scalability to population-level analyses by applying it to a single-cell RNA-seq atlas constructed using publicly available data that includes 200,436 cells derived from 47 patients and accounts for three different regions of colorectal tumors and healthy adjacent tissue. The differences detected by SCORPION between intra- and intertumoral regions are consistent with our understanding of disease progression through the chromosomal instability pathway (CIN) that underlies the majority of all colon cancers[14]. Findings were confirmed in an independent cohort of patient-derived xenografts from left- and right-sided tumors and provide insight into the regulators associated with the phenotypes and the differences in their survival rate.

## Results

### The SCORPION algorithm

SCORPION is an R package that generates through five iterative steps comparable, fully connected, weighted and directed transcriptome-wide gene regulatory networks from single-cell transcriptomic data that are suitable for their use in population-level studies (Fig. 1a). To begin, the highly sparse high-throughput single-cell/nuclei RNA-seq data are coarse-grained by collapsing a $k$ number of the most similar cells identified at the low-dimensional representation of the multidimensional RNA-seq data. This approach reduces sample size while also decreasing data sparsity, allowing us better to capture the strength of the relationship between genes' expression[12].

The second step is to construct three distinct initial unrefined networks, as described in the PANDA algorithm: the co-regulatory network, the cooperativity network and the regulatory network[11]. The co-regulatory network represents co-expression patterns between genes. This network is constructed using correlation analyses over the coarse-grained transcriptomic data. The cooperativity network accounts for the known protein–protein interactions between transcription factors. This information is downloaded from the STRING database. The third network is the unrefined regulatory network that describes the relationship between transcription factors and their target genes through transcription factors footprint motifs found in the promoter region of each gene.

Following the construction of the three networks, a modified version of the Tanimoto similarity designed to account for continuous values is used to generate the availability network ($A_{ij}$), representing the information flow from a gene $j$ to a transcription factor $i$, describing the accumulated evidence for how strongly the transcription factor influences the expression level of that gene, taking into account the behavior of other genes potentially targeted by that transcription factor. In addition, the responsibility network ($R_{ij}$) is generated by computing the similarity between the cooperativity network and the regulatory network. The responsibility represents the information flowing from a transcription factor $i$ to a gene $j$ and captures the accumulated evidence for how strongly the gene $j$ is influenced by the activity of that specific transcription factor, taking into account other potential regulators of gene $j$.

The average of the availability and the responsibility networks is computed in the fourth step, and the regulatory network is updated to include a user-defined proportion ($\alpha = 0.1$ by default) of the information provided by the other two original unrefined networks. The cooperativity and co-regulatory networks are also updated in the fifth step using the new information contained in the updated regulatory network. Steps three to five are repeated iteratively until the Hamming distance between the networks reaches a user-defined threshold (0.001 by default). When convergence is reached, the refined regulatory network is returned as a matrix with transcription factors in the rows and target genes in the columns. The matrix values encode the strength of the relationship between each transcription factor and gene. A more detailed description of all the methodological steps performed in SCORPION is available in Methods.

### Comparison against existing methods

To provide a comparison of how data desparsification in SCORPION would affect downstream network modeling, we tested its performance against other algorithms. To do so, we conducted a systematic comparison of network construction algorithms using BEELINE, an

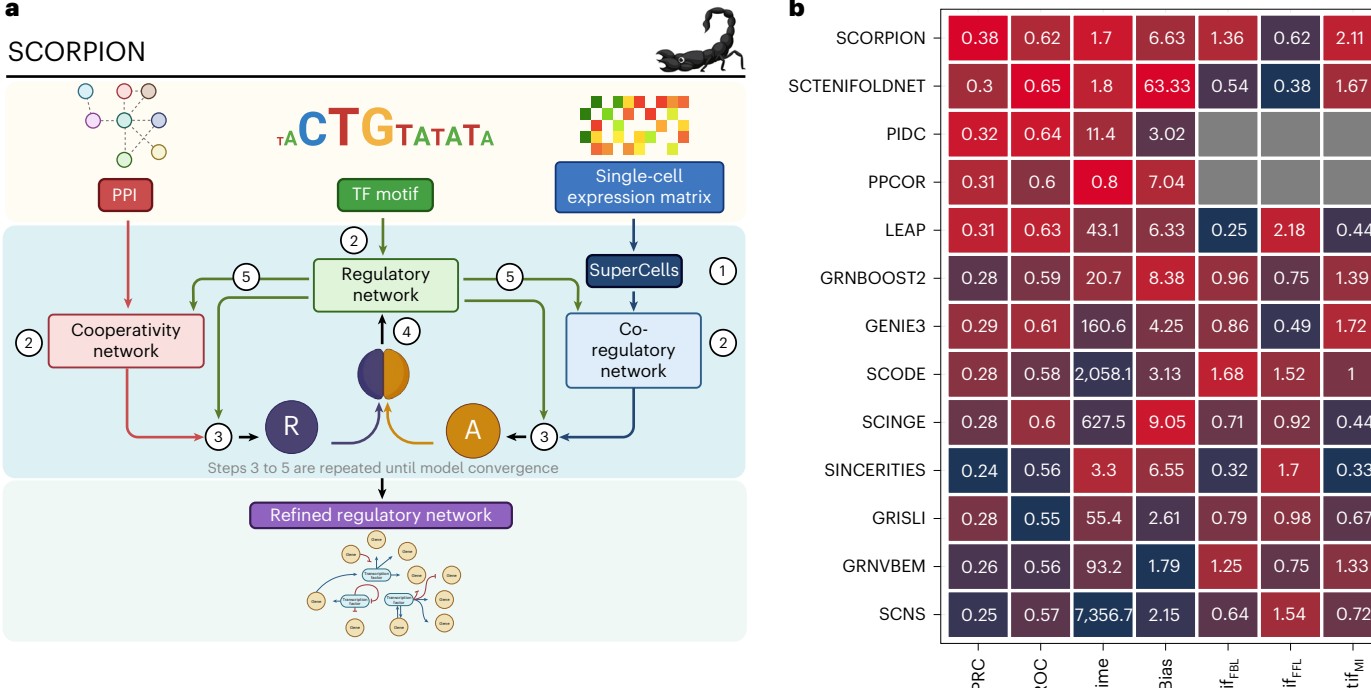

**Fig. 1 | Overview and benchmarking of desparsification with SCORPION.**
**a**, SCORPION uses the PANDA message-passing algorithm to integrate data
from multiple sources, including protein–protein interactions (PPI), single-
cell gene expression and sequence motif data, to predict accurate regulatory
relationships. In five iterative steps, SCORPION generates comparable, fully
connected, weighted and directed transcriptome-wide gene regulatory
networks from single-cell transcriptomic data suitable for use in population-
level studies. TF, transcription factor. **b**, The performance of 13 single-cell gene
regulatory network construction methods was evaluated using BEELINE and
the same curated synthetic dataset. Methods are ranked based on their average
performance across seven different metrics. If the metric was not quantifiable,
gray squares are shown. The performance in each metric is color-coded from
red (best) to blue (worst). Algorithms were ranked based on their average
performance across seven different metrics: AUROC, AUPRC, computing time,
level bias due to expression level, feedback loops (FBL; where some portion (or
all) of a regulatory response is used as input for future gene regulation), feed-
forward loop (FFL; a three-gene pattern composed of two input transcription
factors, one of which regulates the other, both of which jointly regulate a
target gene) and mutual iterations (MI; equally weighted interactions between
regulator–target and vice versa) motif structures identification. AUROC and
AUPRC are described in Methods. The absolute value of the correlation between
the average gene expression for each gene and its corresponding degree in the
network was used to calculate the level bias due to expression level.

evaluation tool designed for this purpose[13]. SCORPION was tested and
compared with 12 different algorithms. Each method's performance
in recovering gene-to-gene relationships was compared with ground-
truth interactions between genes generated using pre-set parameters
without other information than the expression matrix. According to
our findings, SCORPION generates 18.75% more precise (higher preci-
sion) and sensitive (higher recall) gene regulatory networks than other
methods. Furthermore, in our analysis, we found that while PPCOR
and PIDC show similar performance to SCORPION, they are limited
in their ability to evaluate all the regulatory mechanisms expected to
be represented in a gene regulatory network and do not perform well
in transcriptome-wide scenarios (Supplementary Fig. 2). In addition,
when compared with other methods using seven different metrics
related to network construction, SCORPION consistently ranks first
on average (Fig. 1b and Supplementary Table 1).

The curated dataset provided by BEELINE to perform the bench-
mark of the different tools is much simpler than the transcriptome-wide
gene regulatory network required in reality to identify mechanistic
changes in gene regulation that support the observed phenotypes. In
fact, it is known that incorporating prior information on transcription
factor binding into regulatory network reconstruction algorithms
improves predictions of regulation[15]. For that reason, after having
tested the outperformance of SCORPION's desparsification approach
on synthetic data, we chose to apply the complete SCORPION frame-
work—desparsification with SuperCells (this procedure is sometimes

also referred to as meta-cells or (mini) pseudo-bulks)[12] and message
passing between prior regulatory, cooperativity and co-regulatory
networks—directly to curated real datasets and assess the biological
relevance of the generated gene regulatory networks.

## Detection of changes in transcription factor activity
We used two curated real datasets generated using 10x Genomics'
high-throughput single-cell/nuclei RNA-seq technologies to evalu-
ate SCORPION's performance in identifying changes in transcription
factor activity and their impact on target genes. The first dataset was
generated to examine the redundant effect of Hnf4α and Hnf4γ tran-
scription factors in the intestinal epithelium of mice through a double
knockout (DKO) experiment[16]. The second dataset was designed to
investigate the role of over-expressing the DUX4 transcription factor
on human embryonic stem cells (ESCs) during human zygotic genome
activation-like transcription in vitro[17].

For the first dataset, two independent single-cell gene regulatory
networks were built to model the regulatory mechanisms of $Hnf4\alpha\gamma^{WT}$
(wild type, $n = 4{,}100$) and $Hnf4\alpha\gamma^{DKO}$ ($n = 4{,}200$) mouse intestinal
epithelial cells. The $Hnf4\alpha\gamma^{WT}$ network models the regulation of 4,255
genes by 603 transcription factors, while the $Hnf4\alpha\gamma^{DKO}$ network
accounts for the regulation of 3,384 genes by the same amount of
transcription factors as in $Hnf4\alpha\gamma^{WT}$. We used the subnetwork repre-
senting the regulatory mechanisms of the 2,990 genes that overlapped
in both networks for comparison. We focused on the differences in the

edge weights of the Hnf4α and Hnf4γ transcription factors because they represent the changes on the transcription factor's activity over their target genes' expression after perturbation. In both cases, we observed a shift in the weights of the links between the perturbed transcription factors and their target genes (Fig. 2a,e). The paired weight differences were found to be highly significant ($t$-test, $P = 1.1 \times 10^{-85}$), and the direction of the shift ($\hat{\mu}_{Hnf4\alpha} = -0.24$ and $\hat{\mu}_{Hnf4\gamma} = -0.21$) consistent with the perturbation targeted (downregulation) in the cells during the experimental design (Fig. 2b,f).

We identified 221 and 211 large changes (outside the 95% confidence interval, 181 genes shared, Jaccard index 0.819) after the experimental perturbation of *Hnf4α* and *Hnf4γ*, respectively. These changes (Fig. 2c,g) highlight 84 shared genes with decreased activation signal (downregulation) from 114 in *Hnf4α*- perturbed and 95 in *Hnf4γ*-perturbed cells (Jaccard index 0.672), as well as 97 shared genes with increased activation signal (upregulation) from 107 in *Hnf4α*-perturbed and 116 in *Hnf4γ*-perturbed cells (Jaccard index 0.769). The high overlap (81.9%) in the top-most perturbed target genes discovered after the DKO supports the paralog redundant activity of Hnf4α and Hnf4γ in the intestinal epithelium of mice. In addition, in agreement with what the dataset's original authors reported[16], when we performed gene set enrichment analysis (GSEA) using the paired differences between the weights of the link between the transcription factors and their target genes, we found that *Hnf4α* and *Hnf4γ* perturbations have a significant (normalized enrichment score (NES) < 0 and false discovery rate (FDR) < 0.05) impact on reducing the expression of the canonical marker genes associated with enterocyte identity development (Fig. 2d,h and Supplementary Tables 2 and 3).

In addition, to examine SCORPION's performance to integrate multiple sources of information and the impact of using random priors on network construction, we conducted a comprehensive assessment by introducing randomization into the priors data 50 times, each with different seeds. We used the DKO dataset described above to assess how random regulatory priors would affect network learning. Our analysis revealed a pattern: the incorporation of random priors significantly reduced the disparities between the network representing the perturbed sample and the wild-type reference. This was evident in the Spearman correlation coefficient, which increased from 0.88 when using the correct priors to an average of 0.95 with randomized ones. This difference was statistically significant (one-sided $t$-test $P = 2.2 \times 10^{-16}$). In addition, there was a smaller average difference in the edge weights of both Hnf4α (from −0.24 to −0.17 on average; one-sided $t$-test $P = 3.25 \times 10^{-11}$) and Hnf4γ (from −0.21 to −0.17 on average; one-sided $t$-test $P = 2.51 \times 10^{-15}$) transcription factors to their target genes in networks using randomized priors (Supplementary Fig. 3).

For the second dataset, as before, we constructed two independent gene regulatory networks to model the regulatory mechanisms on wild-type human ESCs and the effect of over-expressing (OE) the DUX4 transcription factor in them. The resulting two gene regulatory networks represent the regulatory effect of 622 transcription factors over 13,422 genes in 970 *DUX4*^WT human ESCs and a subset of 55 *DUX4*^OE human ESCs exhibiting the canonical marker genes (*ZSCAN4, DUXA, CCNA1* and *KDM4E*) of 8-C-like cells (Fig. 2I and Supplementary Table 4). When we compared the transcription factor activity of *DUX4* in both networks, we noticed a shift in distribution of the weights of the links before and after the transcription factor was overexpressed (Fig. 2j). In agreement with the experimental design targeted in the human ESCs, we found that the paired differences in the weights of the links between DUX4 and its target genes are significantly ($t$-test, $P < 0.0001$) shifted to the positive side (Fig. 2k), inducing upregulation of its target genes. We found 999 extreme link weight changes outside the 95% confidence interval, which represent 624 and 375 target genes down- and upregulations associated with the overexpression of DUX4 on human ESCs respectively (Fig. 2I). When we performed GSEA using

the paired differences between the weights of the links between DUX4 and its target genes, we found that these are positively associated (NES > 0, $P < 0.05$) with the overexpression of highly expressed genes in 8C-like cells such as *ADD3, ALPG, BCAT1, DPPA3, EXOSC10, HIPK3, NEAT1, ODC1, RBBP6, RBM25, SAMD8, SLC2A3, WDR47* and *ZNF217* (Fig. 2m, Supplementary Table 5).

These findings confirm that SCORPION can detect experimentally targeted changes in transcription factor activity and represent the impact of those changes on the resulting gene regulatory networks. This holds true when comparing two networks. However, as SCORPION networks are refined using a message-passing algorithm, the only difference between the resulting networks is given by the correlation structure provided by the RNA-seq data from single cells/nuclei used to generate the co-regulatory network. This feature, in conjunction with the short time of construction (Fig. 1b), makes SCORPION suitable for the generation of comparable gene regulatory networks in a pipeline scalable to population-level studies targeting the identification of differences in gene regulation. To showcase this feature, we chose to use SCORPION to reconstruct gene regulatory networks for each cell type within each sample in a multi-sample single-cell atlas of colorectal cancer that includes cells from both nearby normal tissue and three distinct tumor regions.

### Reflecting cellular identity and disease status

We generated a multi-sample single-cell RNA-seq atlas containing the transcriptomes of cells from adjacent healthy tissue and three different regions of colorectal tumors, including metastatic, core and border tissue aiming to characterize the regulatory mechanisms driving the development and progression of colorectal cancer. To begin, we gathered single-cell RNA-seq data from five publicly available datasets comprising 303,221 cells derived from 47 donors. After quality control, 200,439 were kept (Fig. 3a–d and Supplementary Table 6). SCORPION was then used to generate a gene regulatory network for each cell type (with at least 30 cells) within each sample included in the atlas after cells were annotated using canonical markers (Supplementary Fig. 1). In total, we generated 560 transcriptome-wide gene regulatory networks that account for the regulatory effect of 622 transcription factors over 17,425 target genes (a total of 10,838,350 links) in each network.

We used the network's indegrees (the sum of the weights from all transcription factors to a gene) to generate a $t$-distributed stochastic neighbor embedding ($t$-SNE) low-dimensional representation of the information contained in the networks. We found that networks of cells of the same type cluster together regardless of tissue of origin (Fig. 3e). This reaffirms the ability of SCORPION to accurately identify the differences in regulatory mechanisms defining cell-type identity across multiple samples.

We chose cells from the core tissue, border tissue and adjacent healthy tissue from four different donors to compare their similarities, to assess the reproducibility of the built gene regulatory networks (Supplementary Fig. 4). We found that, on average, the similarity between the cancer tissue (core and border) is significantly ($t$-test, $P = 3.5 \times 10^{-3}$) higher ($\hat{\mu}_{\hat{\rho}} = 0.945$; Supplementary Fig. 4a) than the one observed when comparing the cancer tissue with the healthy adjacent one ($\hat{\mu}_{\hat{\rho}} = 0.821$; Supplementary Fig. 4b). This outcome confirms our previous findings, in which we were able to reconstruct two gene regulatory networks that represented the control of 15,493 genes through 622 transcription factors in T cells derived from two samples taken from the same benign polyp in a female donor with adenomatous polyposis. Those networks showed a highly positive and significant Spearman correlation coefficient ($\hat{\rho} = 0.931, P = 2.2 \times 10^{-16}$).

### Revealing colorectal cancer progression patterns

One of the most significant advantages of using single-cell/nuclei RNA-seq data is the ability to characterize the molecular mechanisms

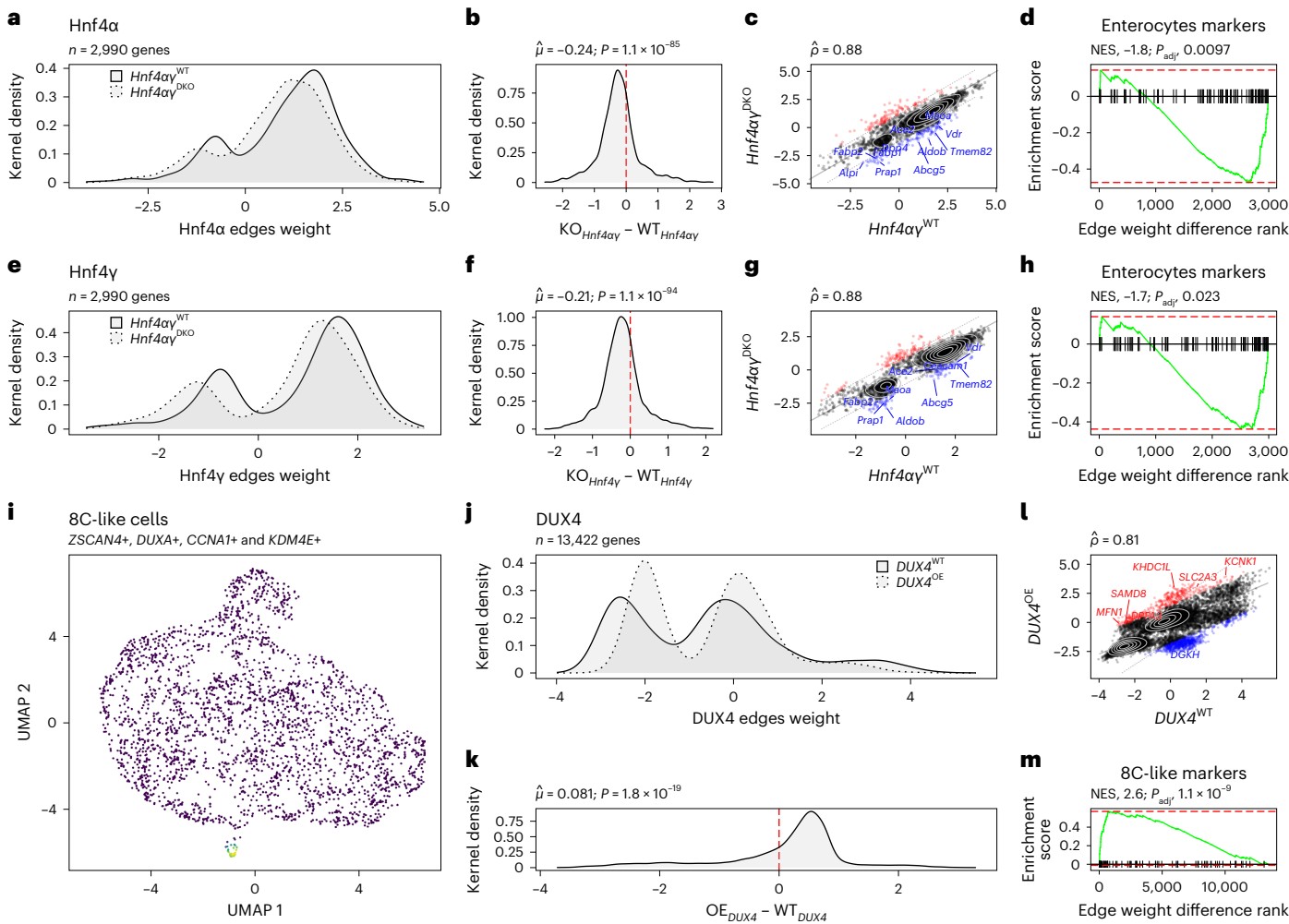

**Fig. 2 | Evaluation of SCORPION's ability to detect changes in transcription factor activity and their impact on target genes. a**, Differences in the distribution of the edge weights for the Hnf4α transcription factor in *Hnf4αγ*^WT^ and *Hnf4αγ*^DKO^ mouse intestinal epithelium cells. **b**, Distribution of the paired weight differences between the edges of the Hnf4α transcription factor ($\hat{\mu}$ and $P$ were calculated using a one-sample two-sided $t$-test). **c**, Spearman correlation ($\hat{\rho}$) of the edge weights for the Hnf4α transcription factor in *Hnf4αγ*^WT^ and *Hnf4αγ*^DKO^ mouse intestinal epithelium cells. Genes outside the 95% confidence interval are color-coded and labeled (in red if upregulated and in blue if downregulated). **d**, GSEA of enterocyte marker genes using the paired differences between the edge weights of the Hnf4α transcription factor (NES and $P_{adj}$ were computed using the GSEA test). **e**, Differences in the distribution of the edge weights for the Hnf4γ transcription factor in *Hnf4αγ*^WT^ and *Hnf4αγ*^DKO^ mouse intestinal epithelium cells. **f**, Distribution of the paired weight differences between the edges of the Hnf4γ transcription factor ($\hat{\mu}$ and $P$ were calculated using a one-sample two-sided $t$-test). **g**, Spearman correlation ($\hat{\rho}$) of the edge weights for the Hnf4γ

transcription factor in *Hnf4αγ*^WT^ and *Hnf4αγ*^DKO^ mouse intestinal epithelium cells. Genes outside the 95% confidence interval are color-coded and labeled (in red if upregulated and in blue if downregulated). **h**, GSEA of the enterocyte marker genes using the paired differences between the edge weights of the Hnf4γ transcription factor (NES and $P_{adj}$ were computed using the GSEA test). **i**, UMAP of human ESCs. 8-cell-like cells are highlighted. **j**, Differences in the distribution of the edge weights for the DUX4 transcription factor in *DUX4*^WT^ and *DUX4*^OE^ human ESCs. **k**, Distribution of the paired weight differences between the edges of the DUX4 transcription factor ($\hat{\mu}$ and $P$ were calculated using a one-sample two-sided $t$-test). **l**, Spearman correlation ($\hat{\rho}$) of the edge weights for the DUX4 transcription factor in *DUX4*^WT^ and *DUX4*^OE^ human ESCs. Genes outside the 95% confidence interval are color-coded and labeled (in red if upregulated and in blue if downregulated). **m**, GSEA of the 8-C-like cell marker genes using the paired differences between the edge weights of the DUX4 transcription factor (NES and $P_{adj}$ were computed using the GSEA test).

underlying disease at the cell-type-specific level. Because colorectal cancer is an epithelial cancer, we decided to focus on the molecular mechanisms that drive disease progression in epithelial cells. We selected the 149 single-cell gene regulatory networks generated for this cell type among the four tissues (healthy $n$ = 42, border $n$ = 9, core $n$ = 94 and metastasis $n$ = 4), and used linear regression to investigate each of the 9,532,150 links between 622 transcription factors and 15,325 target genes aiming to identify linear patterns of up- or downregulation across these links. Our reasoning was that healthy adjacent tissue (encoded as 1) is transitionally transformed into malignant tissue along the border (encoded as 2), and disease signals will be increased in the tumor's core (encoded as 3)

and metastatic tissue (encoded as 4). We calculated a $\beta$ coefficient and associated adjusted for multiple testing $P$ value for each link (Fig. 4a). We found 5,202,588 links with a absolute value of $\beta$ greater than 0 and an FDR less than 0.05. We treated these $\beta$ coefficients as weights in the generated network representing colorectal cancer progression (Fig. 5 and Supplementary Table 7).

We found that some of the identified interactions have directions that are consistent with previously reported oncogenic transformation patterns necessary for the growth and development of colorectal tumors (Fig. 4a). For example, upregulation of *EGR2* is required for colon cancer stem cells survival and tumor growth[18], upregulation of *HDAC5* promotes colorectal cancer cell proliferation[19], upregulation

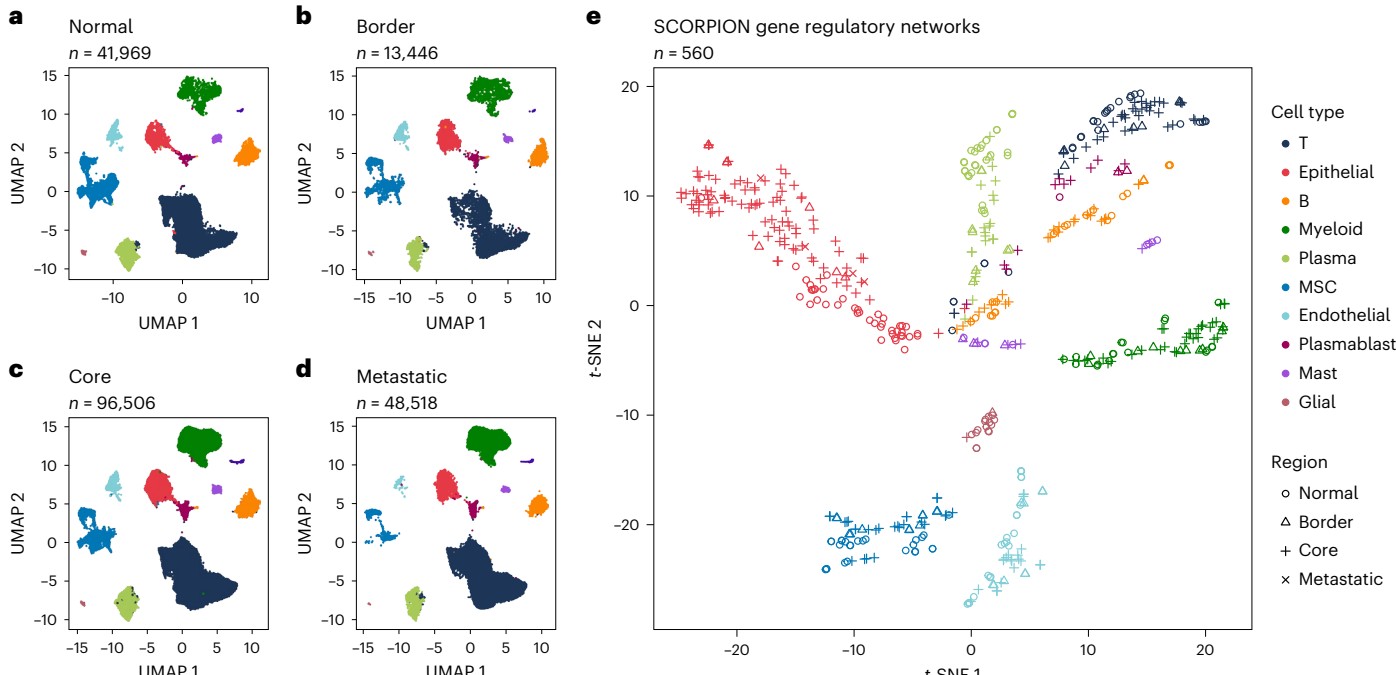

**Fig. 3 | Low-dimensional representation of transcriptomes and gene regulatory networks from colorectal cancer and adjacent healthy tissue.** **a**, UMAP of cells from healthy adjacent tissue. **b**, UMAP of cells from tumor border tissue. **c**, UMAP of cells from tumor core tissue. **d**, UMAP of cells from liver metastatic tissue. **e**, t-SNE of gene regulatory networks from colorectal cancer and adjacent healthy tissue generated by SCORPION. MSC, mesenchymal stem cell.

of *SP1* activates the Wnt/β catenin pathway in colorectal cancer[20], upregulation of *CCND2* in conjunction with *JAK2* and *STAT3* promotes colorectal cancer stem cell persistence[21], upregulation of *NANOG* modulates stemness in human colorectal cancer[22], upregulation of *ADGRG1* promotes proliferation of colorectal cancer cells and enhances metastasis via the epithelial-to-mesenchymal transition[23]. Examples, where edge weights are reduced through tumor progression include the inhibition of the epithelial-to-mesenchymal transition during cancer metastasis by *HDAC2*[24], and the tumor-suppressing role in colorectal cancer by *HOXD8* that act as an apoptotic inducer[25].

To identify the major drivers of colorectal cancer progression, we calculated transcription factor overall association as the (outdegree) sum of all the β coefficients for each transcription factor to its target genes. We found that the top ten most associated transcription factors across colorectal cancer development are *ZNF770, SP1, SP2, SP3, PATZ1, MAZ, PAX5, KLF15, WT1* and *KLF3*. Among these, *SP1, WT1, PAX5* and *KLF3* are known to be associated with transcriptional misregulation in cancer (hyper-geometric test, Kyoto Encyclopedia of Genes and Genomes (KEGG) database, Odds Ratio = 70.22, FDR < 0.0001). In contrast, the top ten associated transcription factors with reduced outdegrees throughout tumor progression are *ZNF146, ZNF490, BCL6B, SOX11, ZBED1, ZNF250, GLIS1, ZNF586, HOMEZ* and *VSX2* (Fig. 4b).

We also calculated the network's indegrees by aggregating the regulation of all transcription factors over a target gene. We used this vector of aggregated weights to represent the rate of change of each gene during disease progression, by performing linear regression on the indegrees. We then evaluated gene set enrichment using the hallmarks of cancer as ref. 26. Out of the 50 hallmarks, we found 11 significantly (FDR < 0.05) perturbed. Mitotic spindle, Hedgehog signaling, and Wnt/β catenin signaling were among the six hallmarks found to be upregulated (NES > 0). These three characteristics are part of a well-known colorectal cancer pathway known as the CIN pathway. The CIN pathway is linked to an increase in genomic instability, which is critical for the development of colorectal cancer. CIN is also the most

common cause of colorectal cancer[27]. In addition, we found that the c-Myc pathway in the epithelial cells of the tumor's core and metastasis regions was significantly downregulated (NES < 0). This is in line with earlier reports suggesting that low c-*MYC* levels enable cancer cells to survive in the presence of low levels of oxygen and glucose, which are characteristic of the tumor's core[28].

Overall, we found that the regulatory patterns represented in the gene regulatory networks generated by SCORPION to characterize the progression of colorectal cancer in epithelial cells strongly agree with our understanding of the disease's progression. These high-quality data with unparalleled resolution due to the use of single-cell RNA-seq show that SCORPION is suited for the construction of comparable gene regulatory networks to support population-level comparisons aimed at identifying differences in gene regulation.

We next wanted to demonstrate the potential of SCORPION to identify differences in gene regulatory networks between conditions. There are four accepted consensus molecular categories for colorectal cancer, CMS1 (microsatellite instability immune), CMS2 (canonical), CMS3 (metabolic) and CMS4 (mesenchymal), which were determined based on the tumor's composition and mutational status[29]. A genetic cascade of changes causes the normal colonic epithelium to first become an adenoma and subsequently an adenocarcinoma as colorectal cancer progresses. For this reason, it is essential to first comprehend and give priority to the regulatory mechanisms of malignant epithelial cells to develop pharmacological options for patients. It is well recognized that the origin, phenotype and prognosis of cancer arising from different sides of patients' intestines vary. Whereas differences in tumor composition and differential gene expression at the single-cell atlas level have been reported before[30], a differential gene regulatory network analysis aiming to identify regulatory drivers of the differences has not been conducted at this level of resolution. We therefore chose to contrast the regulatory processes defining colorectal tumors arising on the left (splenic flexure, sigmoid colon, descending colon and rectum) and right (cecum, appendix, ascending colon and hepatic flexure) sides of the patients' intestines.

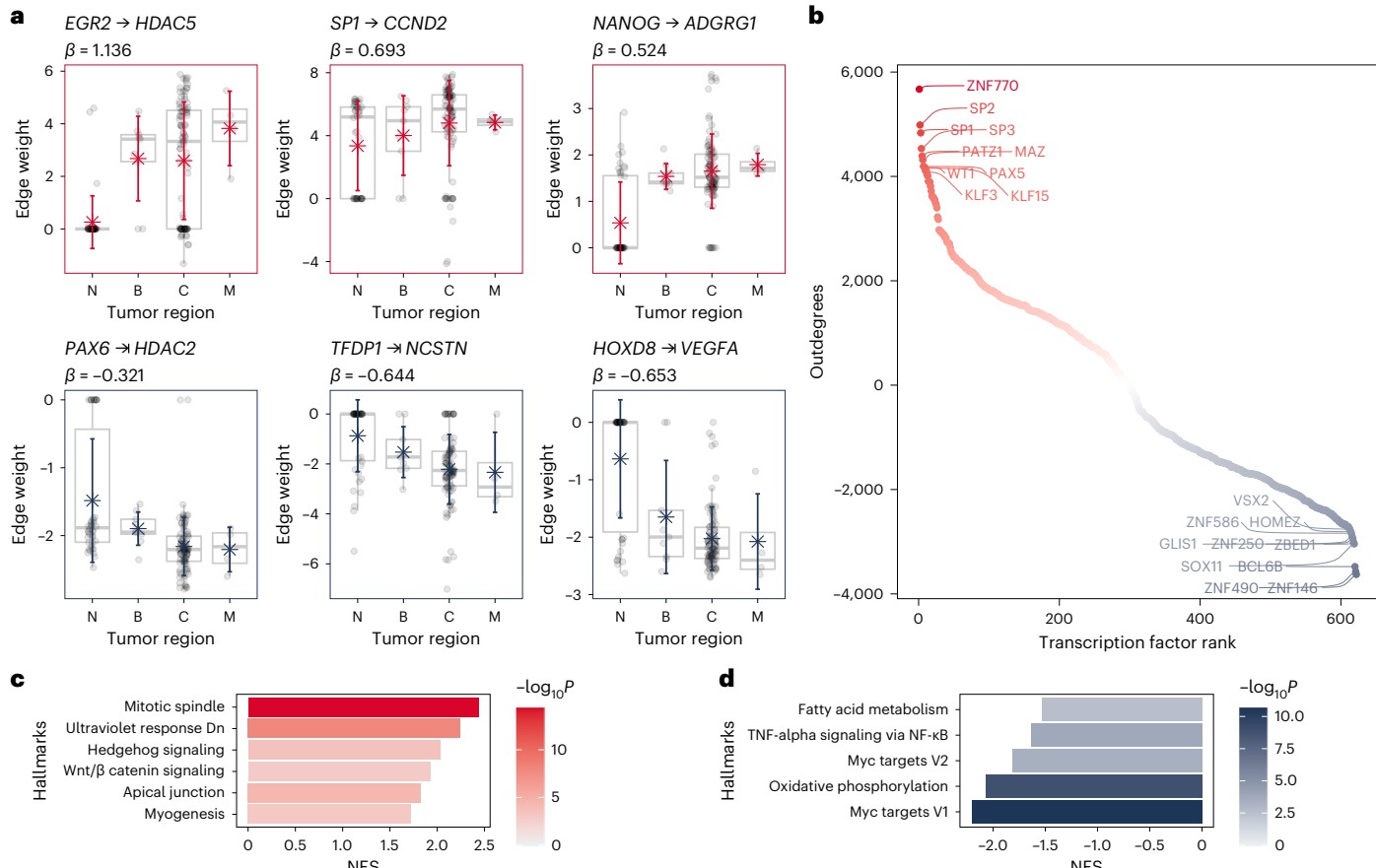

**Fig. 4 | Differential network analysis of epithelial cells during colorectal cancer progression.** N, adjacent normal tissues, B, border of the tumor, C, core of the tumor, M, liver metastases. We analyzed *n* = 149 biologically independent samples, with sample sizes for each condition as follows: N = 42, B = 9, C = 94, M = 4. In the boxplots, the line within the box represents the median. The box itself extends from the median ± 1.5 times the interquartile range (IQR). Whiskers indicate the 5th and 95th percentiles, and individual sample values are represented as dots. **a**, Examples of significant interactions between transcription factors and target genes linearly increasing or decreasing during colorectal cancer progression (*β* coefficient computed using ordinary least squares). **b**, Ranked list of transcription factors based on the transcription factor activity in the gene regulatory networks illustrating the progression of colorectal cancer. **c**, Significantly upregulated hallmarks found in the gene regulatory network illustrating the progression of colorectal cancer, ranked by NES (NES and $P_{adj}$ were computed using the GSEA test). **d**, Downregulated hallmarks found in the gene regulatory networks illustrating the progression of colorectal cancer, ranked by NES (NES and $P_{adj}$ were computed using the GSEA test).

To determine the drivers of regulatory differences across epithelial cells from the core of 11 right-sided and 22 left-sided colorectal tumors (Methods), we computed transcription factor targeting (outdegree) for each of the 622 transcription factors in each network independently (Fig. 6a). After comparing the two groups, we found 118 transcription factors with enhanced activity in right-sided colorectal cancer in contrast with the 287 found with enhanced targeting in left-sided colorectal cancer (Fig. 6b). Among the top ten more active transcription factors in left-sided colorectal cancer (Fig. 6c) we found a significant enrichment of transcription factors associated with unfolded protein response (*NFYA* and *CEBPG*, hypergeometric test, FDR < 0.01). In right-sided ones (Fig. 6d), we found an enrichment of transcription factors associated with tumor necrosis factor (*TNF*) signaling via nuclear factor kappa B (*NF-κB; KLF9, NFKB1* and *NFKB2*, hypergeometric test, FDR < 0.001). A thorough examination of the unfolded protein response and the NF-κB signaling pathways in colorectal cancer has previously been reported[31]. We found that the most significant drivers of the differences between left-sided and right-sided colorectal cancer found in our analysis are *ZNF350* (*t*-test, FDR = 0.024) and *NFKB2* (*t*-test, FDR = 0.032) respectively.

When these two patterns are combined, they are consistent with the significantly worse survival rate of patients with right-sided

colorectal malignancies[32]. The methylation of the *ZNF350* transcription factor's promoter region, which causes its downregulation, is known to stimulate colon cancer cell migration[33]. In addition, overexpression of *NFKB2* is a known prognostic marker of poor survival in colorectal cancer[34]. To cross-validate these relationships, we first compared the averaged survival rates based on *NFKB2* expression of patients with primary tumors in the cecum, appendix, ascending colon, hepatic flexure, splenic flexure, sigmoid colon, descending colon and rectum from the The Cancer Genome Atlas (TCGA) colon adenocarcinoma (COAD) and TCGA rectum adenocarcinoma (READ) projects[35]. We confirmed the association between the level of *NFKB2* expression and the average survival rate of the patients (log-rank test, *P* = 0.042; Fig. 6e). Following that, we compared the levels of expression of the two transcription factors in primary colorectal tumors on the left and right sides of the intestine. We found that, in both cases, the patterns identified by SCORPION and represented in the gene regulatory networks are consistent in directionality and significance with the level of expression observed in the primary tumors from the TCGA data (left panels in Fig. 6f,g).

To further cross-validate our findings and assess the reliability of this pattern in a smaller population, we compared the expression levels of both transcription factors in a new dataset of 15 patient-derived

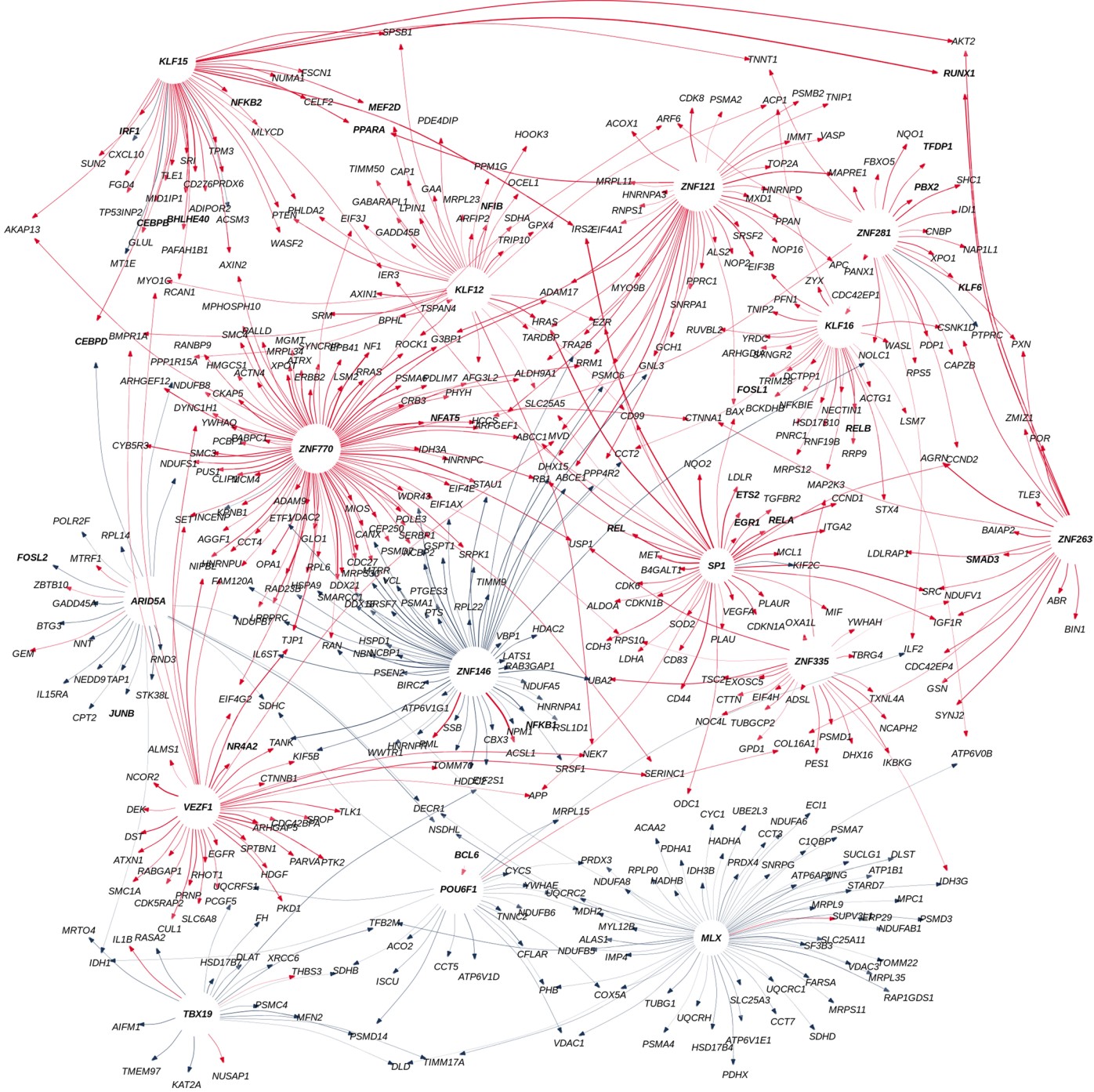

**Fig. 5 | Gene regulatory network illustrating the progression of colorectal cancer.** Transcription factors with the highest activities up- or downregulated are shown in bold letters. The graph's edges are color-coded in red for upregulated and blue for downregulated interactions. Arrows represent the directionality of the regulatory mechanism.

xenograft models (PDXs; Methods) generated by us (Supplementary Table 8). Nine samples were from right-sided and six from left-sided colorectal tumors. Here, as before with the TCGA data, we demonstrated that the patterns identified by SCORPION and represented in the gene regulatory networks are consistent in both directionality and significance with the level of expression observed (right panels in Fig. 6f,g).

These findings highlight SCORPION's ability to identify not only intratumoral characteristics affecting patient survival but also novel biomarkers and appropriate targets for developing pharmacological options for patients.

## Discussion

The use of data other than gene expression distinguishes SCORPION from most other methodologies and allows for the modeling of known perturbations of protein–protein interactions and transcription factor binding patterns. Compared with other algorithms that do incorporate prior information on transcription factors, such as SCENIC[36] and SCIRA[37], SCORPION uses the information about the motif footprints during the construction of the network and not only to characterize the activity of the transcription factors. Furthermore, unlike SCENIC, SCORPION employs an association metric ($\mathcal{Z}$ scores) with a defined underlying distribution ($\mathcal{N}$) that facilitate the comparison of weights

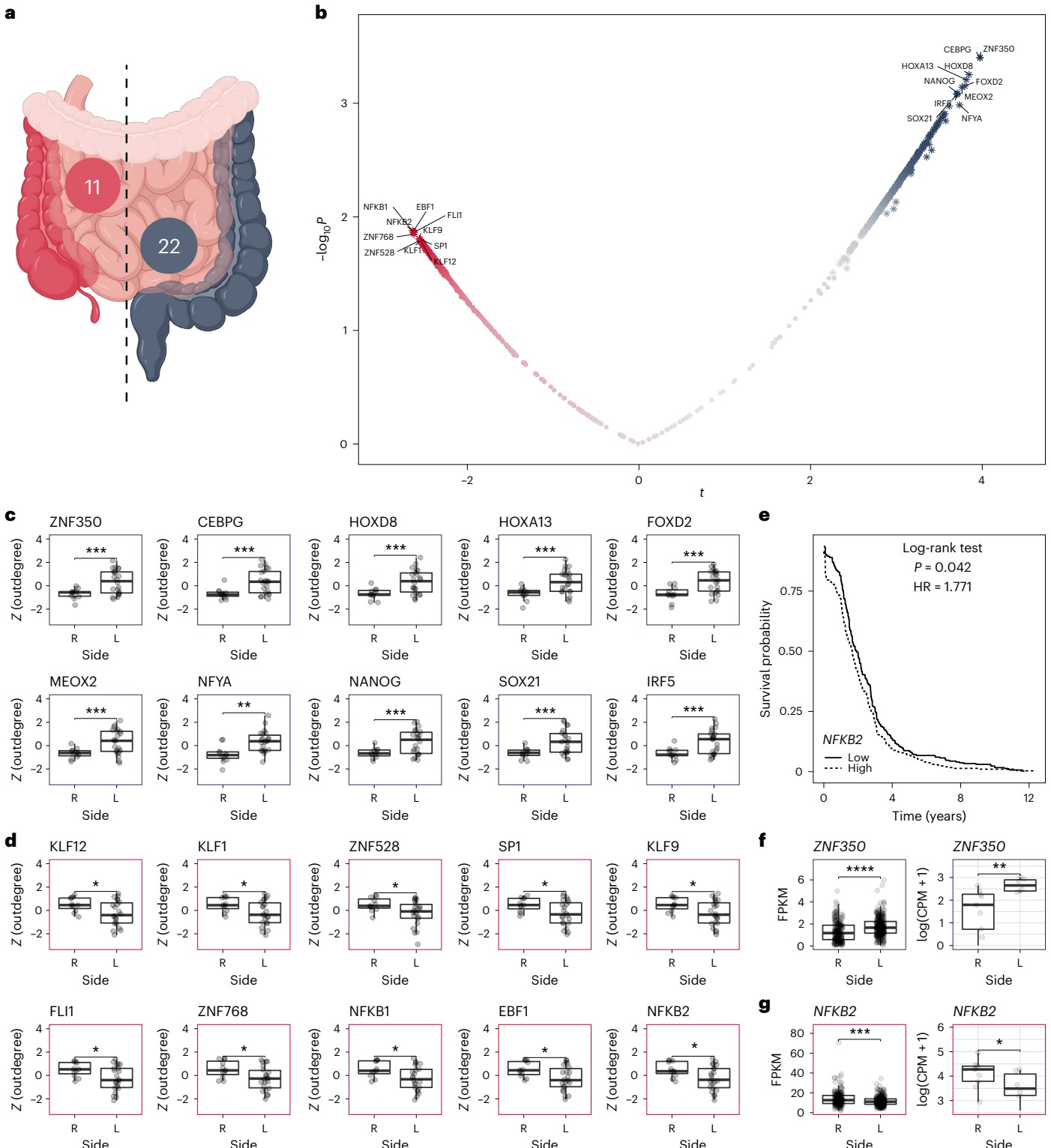

**Fig. 6 | Regulatory differences between right-sided and left-sided colorectal cancer epithelial cells. a**, Diagram illustrating the left and right sides of the intestines, with the respective number of samples for each group. **b**, Volcano plot showing differences in transcription factor activity between right-sided and left-sided colorectal cancer epithelial cells. **c**, Top 10 most active transcription factors identified in epithelial cells from left-sided colorectal cancer based on $n = 33$ biologically independent samples. The dataset includes 22 samples from the left side (L) and 11 from the right side (R). **d**, Top 10 most active transcription factors identified in epithelial cells from right-sided colorectal cancer based on $n = 33$ biologically independent samples. The dataset includes 22 samples from the left side (L) and 11 from the right side (R). **e**, Differences in patient survival rates based

on *NFKB2* expression in patients with primary colorectal cancer. **f**, Consistent differences in gene expression for the ZND350 transcription factor in the TCGA data and our own dataset. **g**, Consistent differences in gene expression for the NFKB2 transcription factor in two independent patient cohorts. Expression levels are reported in fragments per kilobase of transcript per million mapped reads (FPKM) and counts per million (CPM), respectively. In the boxplots, the line within the box represents the median, and the box extends from the median $\pm$ 1.5 times the IQR. Whiskers indicate the 5th and 95th percentiles, and individual sample values are represented as dots. P values were calculated using a two-sided *t*-test: \*$P \le 0.05$, \*\*$P \le 0.01$, \*\*\*$P \le 0.001$, \*\*\*\*$P \le 0.0001$.

across experiments and allowed us to identify edges associated with colorectal cancer progression, and, like SCIRA, SCORPION allows for the quantification of the activity of undetected transcription factors, which is common in high-throughput single-cell transcriptomic data but to our knowledge not possible with SCENIC.

SCORPION also offers notable computational improvements compared with other gene regulatory network construction tools. By default, it utilizes sparse matrices, resulting in reduced memory usage and faster matrix multiplications. In addition, it incorporates truncated principal components for the desparsification step, further enhancing computational efficiency. Furthermore, SCORPION is readily available on multiple platforms through the CRAN repositories, simplifying its installation and use on various operating systems. However, there are certain limitations associated with the use of SCORPION. These include an additional step required to gather prior information on transcription factor motif binding and protein–protein interactions, unlike methods relying solely on transcriptomic data. Moreover, SCORPION necessitates sufficient sequencing depth to ensure robust correlation coefficients; a high number of dropouts or numerous unique cells with distinct phenotypes may result in less accurate networks.

Finally, by constructing precise and highly comparable gene regulatory networks for each sample, SCORPION enables the use of the same statistical techniques that consider population heterogeneity and are widely used in other areas of genomic data analysis. These methods include, but are not limited to, clustering based on sample similarity, dimensionality reduction and differential analysis. We anticipate that SCORPION will be used not only to characterize molecular mechanisms driving phenotypes but also to investigate a wide range of important questions in precision medicine, health and biomedical research now that gene regulatory network perturbations have been shown to be effective at reproducing experimental results[38].

## Methods

### Statistics and reproducibility

This study primarily relies on extensive publicly available datasets. In this context, no statistical method was employed to predefine the sample size, and, after quality control, all data were included in the analyses without exclusion. The experiments were not randomized, and the investigators were not blinded to allocation during both experiments and outcome assessment. All of the data and code required to replicate the analysis as well as the figures and tables are available at https://github.com/dosorio/SCORPION.

### Enhanced details on SCORPION method

SCORPION is an R package that generates through five iterative steps comparable, fully connected, weighted and directed transcriptome-wide gene regulatory networks from single-cell transcriptomic data that are suitable for their use in population-level studies (Fig. 1a). SCORPION uses PANDA's message-passing algorithm to model gene regulatory networks. This method incorporates three input data types—potential protein–protein interactions between transcription factors, an initial estimate of potential transcription factor binding to promoter regions, and co-expression signals derived from transcriptomic data. It then models regulatory interactions through an iterative message-passing process, where it assigns greater significance to connections (edges) between a regulator and a target gene when there is substantial agreement in targeting patterns of regulators that may cooperate in regulating their target genes, as well as in co-expression patterns of these target genes.

The PANDA algorithm starts by creating initial networks for different data types. It then facilitates the exchange of messages between these networks, updating edge values in iterative message-passing steps. This message passing occurs in two steps: estimating and updating the regulatory network, and estimating and updating the protein–protein interaction and gene co-expression networks (the latter modeled with Pearson correlation). PANDA incorporates

protein interactions to predict the responsibility of regulatory relationships. It assumes that transcription factor proteins often collaborate in complexes to regulate genes. The algorithm combines the regulatory network with a protein cooperativity network to predict the responsibility of an edge from a transcription factor to a gene. Co-regulation is employed to predict the availability of regulatory relationships. Genes that share binding motifs for the same transcription factors in their promoter regions are more likely to be co-regulated than genes that do not. The algorithm combines information from the regulatory network with a co-regulation network to predict the availability of a target gene to a specific transcription factor. PANDA then uses the average of the responsibility and availability values to update the initial regulatory network with information learned from the protein–protein interaction and co-expression data. This updated network is then used for further iterations. The algorithm's convergence is determined by calculating the Hamming distance between the current and estimated network. The algorithm also integrates information from the updated regulatory network into co-regulation and protein cooperativity networks. For more information regarding the PANDA algorithm, refer to ref. 11.

Within SCORPION, the process starts with the highly sparse high-throughput single-cell/nuclei RNA-seq data, which is subsequently coarse-grained by collapsing a $k$ number of the most similar cells identified at the low-dimensional representation of the multidimensional RNA-seq data. This approach reduces sample size while also decreasing data sparsity, allowing us better to capture the strength of the relationship between gene expression levels[12].

The second step is to construct three distinct initial unrefined networks: a co-regulatory network consisting of co-expression patterns between genes, a protein cooperativity network and the regulatory network ($W^{(0)}$)[11]. The co-regulatory network is computed using Pearson correlation (as in the original PANDA algorithm) on the coarse-grained expression profiles. The cooperative network accounts for known protein–protein interactions between transcription factors. This information is downloaded from the STRING database[39]. The third network is the unrefined regulatory network that describes potential binding of transcription factors to promoter regions. This can, for example, be based by matching transcription factors footprint motifs to the promoter region of each gene[40]. SCORPION then applies PANDA to these three networks to infer interactions between transcription factors and their target genes, for individual super/meta-cells.

After constructing the three unrefined networks, SCORPION employs the similarity metric used in PANDA—a modified version of the Tanimoto similarity that allows to incorporate continuous values. This modified version is described by equation (1), where $x$ and $y$ denote vectors of values that have been normalized to $z$-score units. This similarity metric is used to determine the agreement between the data represented by multiple networks using a heuristically defined similarity score.

$$T_Z = \frac{\sum_i x_i y_i}{\sqrt{\sum_i x_i^2 + \sum_i y_i^2 - |\sum_i x_i y_i|}} \tag{1}$$

Then, the availability network $A_{ij} = T_Z\left(W_{i.}^{(t)}, C_j^{(t)}\right)$ is generated, representing the information flow from a transcription factor $i$ to a gene $j$, using the accumulated evidence for how strongly the transcription factor influences the expression level of that gene $\left(W_{i.}^{(t)}\right)$, taking into account the behavior of other genes potentially targeted by that transcription factor $\left(C_j^{(t)}\right)$. In addition, the responsibility network $R_{ij} = T_Z\left(P_{i.}^{(t)}, W_j^{(t)}\right)$ is generated by computing the similarity between the cooperativity network and the regulatory network. The responsibility represents the information flowing from a transcription factor $i$ to a gene $j$ and captures the accumulated evidence for how strongly the gene $j$ is influenced by the activity of that specific transcription factor $\left(W_j^{(t)}\right)$, taking into account other potential regulators $\left(P_j^{(t)}\right)$ of gene $j$.

The average $\left(\widetilde{W}_{ij}^{(t)} = 0.5A_{ij}^{(t)} + 0.5R_{ij}^{(t)}\right)$ of the availability and the responsibility networks is computed in the fourth step, and the regulatory network is updated $\left(\widetilde{W}_{ij}^{(t+1)} = (1-\alpha)W_{ij}^{(t)} + \alpha\widetilde{W}_{ij}^{(t)}\right)$ to include a user-defined proportion ($\alpha = 0.1$ by default) of the information provided by the other two unrefined networks. The cooperativity and co-regulatory networks are also updated in the fifth step using the new information contained in the updated regulatory network. Steps three to five are repeated iteratively ($t$) until the Hamming distance between the networks reaches a user-defined threshold (0.001 by default). When convergence is reached, the refined regulatory network is returned as a matrix with transcription factors in the rows and target genes in the columns. The matrix values encode the strength of the relationship between each transcription factor and gene.

### Prior network generation

To generate the unrefined regulatory networks that serve as prior for the message-passing algorithm, we downloaded the promoter region coordinates for each gene from ENSEMBL. We then used TABIX to retrieve the motif footprints and associated MOODS match scores located within 1,000-bp before the transcription start site of each gene from ref. 40. When multiple matches of the same transcription factor footprints were found, the highest value was retained for the study. The data on transcription factor protein–protein interactions and their associated scores were obtained from the STRING database version 11.5[39].

### Synthetic data benchmarking

BEELINE was used to conduct a systematic evaluation of cutting-edge algorithms for inferring single-cell gene regulatory networks[13]. We used SCORPION and 12 other single-cell gene regulatory network inference algorithms on the GSD dataset, which is the largest dataset included in BEELINE and was generated from a curated Boolean model[41]. These techniques include: GENIE3, GRISLI, GRNBOOST2, GRNVBEM, LEAP, PIDC, PPCOR, SCINGE, SCNS, SCODE, SCTENIFOLDNET and SINCERITIES; SCRIBE was excluded from the comparison owing to compatibility issues. We processed the dataset using BEELINE's uniform pipeline, which includes four steps: (1) data pre-processing, (2) docker container generation for SCORPION and the other 12 algorithms mentioned above, (3) parameter estimation, and (4) post-processing and evaluation. No information on transcription factor–target relationships was provided to any of the algorithms we benchmarked SCORPION against throughout the analysis. We compared algorithms based on their average performance across seven different metrics: area under the receiver operating characteristic (AUROC), area under the precision–recall curve (AUPRC), computing time, level bias due to expression level, feedback loops (where some portion (or all) of a regulatory response is used as input for future gene regulation), feed-forward loop (a three-gene pattern composed of two input transcription factors, one of which regulates the other, both of which jointly regulate a target gene) and mutual iterations (equally weighted interactions between regulator–target and vice versa) motif structures identification. AUROC portrays a tested algorithm's performance by presenting the trade-off between true-positive rate TP/(TP + FN) and false-positive rate FP/(FP + TN) across different decision thresholds. AUPRC represents the area under the precision TP/(TP + FP)–recall TP/(TP + FN) curve computed for different decision thresholds between 1 and 0 using, where $P_i$ and $R_i$ are the precision and recall at the $i$th threshold. TP denotes true positive, TN denotes true negative, FP denotes false positive and FN denotes false negative. The absolute value of the correlation between the average gene expression for each gene and its corresponding degree in the network was used to calculate the level bias due to expression level.

### Curated scRNA-seq benchmark

Count matrices for both experiments and conditions were downloaded from the Gene Expression Omnibus (GEO) database with accession numbers GSM3477499, GSM347750, GSM5694433 and GSM5694434. Data were loaded into R using the build-in functions included in Seurat for this purpose[42]. Two networks (one for the WT sample and one for the DKO) were built for the Hnf4αγ experiment using SCORPION (under default parameters). The study was restricted to genes expressed in at least 5% of the cells in each sample. For the *DUX4* experiment, datasets were subject to quality control and integrated using Harmony[43]. Low-dimensional representations and clustering of the data were generated using the top five dimensions returned by Harmony. 8-C-like cells were annotated based on the expression of *ZSCAN4*, *DUXA*, *CCNA1* and *KDM4E* genes using the Nebulosa package[44]. All cells from the WT sample were used to build a gene regulatory network that represented this group (under default parameters). Cells exhibiting the 8C-like markers in the *DUX4* overexpression group were used to generate a gene regulatory network representing them. The study was restricted to genes expressed in at least 5% of the cells in both samples. The information in the rows of the network representing the transcription factor of interest for each sample was contrasted to compare transcription factor activities among samples. The residuals of the linear model trained over the data in each case were used to assess the differences in the activity of the transcription factor over each gene. The residuals of the linear model and the marker genes provided by the PanglaoDB database were used to perform GSEA. Additional markers of the 8C-like cells were defined by differential expression using the Wilcoxon rank sum test after comparing the cluster expressing the known marker genes against all other cells.

### Colorectal cancer scRNA-seq atlas construction

We collected multiple publicly available single-cell RNA-seq count matrices for human healthy adjacent tissue and different regions of colorectal tumors (see 'Data availability'). Datasets were loaded into R and combined into a single 'Seurat' object[42]. Following that, data were subjected to quality control, with only cells with a library size of at least 1,000 counts and falling within the 95% confidence interval of the prediction of the mitochondrial content ratio and detected genes in proportion to the cell's library size being kept. We also removed all cells with mitochondrial proportions greater than 10% (ref. 45). We then used Seurat's default functions and parameters to normalize, scale and reduce the dimensionality of the data using principal component analysis. Harmony was used for data integration[43]. The top 50 dimensions returned by Harmony were used to generate the uniform manifold approximation and projection (UMAP) projections of the data. Cell clustering was carried out using Seurat's built-in functions, default resolution and Harmony embedding as the source for the nearest-neighbor network construction. Clusters were annotated using Nebulosa[44] and the canonical markers provided by ref. 46.

### Colorectal cancer gene regulatory network atlas construction

Using SCORPION under default parameters, we built a gene regulatory network for each cell type within each sample having at least 30 cells in the constructed colorectal cancer single-cell RNA-seq atlas. We only included genes that were expressed in more than five cells in each sub-sample. For each network, the sum of the activity of all transcription factors over each gene (indegrees) was computed and assembled in a matrix. We used principal component analysis to reduce the dimensionality of the data to the top 50 principal components. We used this data as input for the generation of the $t$-SNE projection. Networks are color-coded as their respective cell type in the single-cell RNA-seq atlas.

### Modeling Colorectal cancer progression patterns

We selected the gene regulatory networks representing the epithelial cells (*EPCAM*+) of the different tumor regions (border, core and metastatic) and the healthy adjacent tissue. We modeled each edge weight representing the transcription factor–target gene interaction across the four different stages. We computed a $\beta$ coefficient representing the

average rate of change across each stage for each edge. The significance of the $\beta$ coefficient was assigned using the $F$ distribution. Adjustment of the $P$ values for multiple testing was performed using FDR.

## Comparing right- and left-sided tumor gene regulatory networks

We selected the generated gene regulatory networks representing the epithelial cells from right- and left-sided tumors. For each network, we computed the (outdegrees) sum of all the activities for each transcription factor over all the genes. We then compared the outdegrees using the t.test function included in the Rfast package. $P$ values were adjusted for multiple testing using FDR.

## PDX establishment

The University of Texas at Austin and The University of Colorado Institutional Animal Care and Use Committee approved all animal procedures. The PDX models were derived in the same manner as described previously[47]. Briefly, 2–3 mm pieces of colorectal tumor sample collected under Institutional Review Board-approved protocol at the University of Texas Dell Medical School and the University of Colorado Cancer Center were engrafted onto the right and left hind flanks of 5-to-6-week-old Nu/Nu mice (Envigo). Tumor volumes were measured by digital calipers every 3 to 4 days and were calculated by $V = 0.52 \times (\text{length} \times \text{width}^2)$. Mice were killed when tumors reached 1.5 cm$^3$ to further propagate the PDX model to the next generation or frozen as a viable tumor (RPMI media containing 10% FBS and 10% DMSO as a freezing media) in $LN_2$ for long term storage. At the time of tumor collection, a portion of the tumor was flash frozen in $LN_2$ for RNA isolation and sequencing. RNA was isolated using PureLink kit (Thermo Fisher) following the manufacturer's protocol. When the tumor specimen was abundant enough, a portion of the tissue sample was flash frozen, and RNA was isolated directly from that tissue. The RNA sample was outsourced to Novogene US subsidiary and UC Davis Sequencing Center, Sacramento, CA for RNA quality control, library preparation and sequencing. Data obtained from Novogene as FASTQ files were subjected to further analysis.

## RNA-seq expression quantification

Gene expression from FASTQ files was quantified using STAR. The computed values for each PDX were loaded into R to generate the expression matrix. The t.test function was used to compare the expression levels of both (ZNF350 and NFKB2) transcription factors.

## Reporting summary

Further information on research design is available in the Nature Portfolio Reporting Summary linked to this article.

## Data availability

The following datasets were used to construct the colorectal cancer single-cell RNA-seq atlas used in this study: ref. 48, accessible through GEO GSE132465 and GSE144735; ref. 46, accessible through GSA HRA000979; ref. 49, accessible through ArrayExpress E-MTAB-8107; and ref. 50, accessible through GEO GSE178318. Gene expression quantification of the patient-derived xenografts generated for this study is available as Supplementary Table 8. All the generated networks, as well as the unrefined networks for human (hg38) and mice (mm10) genes, are available as independent files at https://doi.org/10.5281/zenodo.10515946 (ref. 51). Source data are provided with this paper.

## Code availability

The SCORPION multi-platform stable package is available at https://CRAN.R-project.org/package=SCORPION. Versions under development are available at https://github.com/kuijjerlab/SCORPION and https://doi.org/10.5281/zenodo.10515946 (ref. 51).

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

## Acknowledgements

This work was supported by the Biomedical Research Computing Facility of the University of Texas at Austin. Figure 6a was created with BioRender.com. This work was funded by the European Union's Horizon 2020 research and innovation program under the Marie Sklodowska-Curie grant agreement 801133 (to D.O.), the National Institutes of Health grant GM133658 (to S.S.Y.), the Komen Foundation grant CCR19609287 (to S.S.Y.), the Norwegian Research Council, Helse Sør-Øst and the University of Oslo through the Centre for Molecular Medicine Norway grant 187615 (to M.L.K.), the Norwegian Research Council grant 313932 (to M.L.K.), the Norwegian Cancer Society grants 214871 and 273592 (to M.L.K.) and the Cancer Prevention and Research Institute of Texas (CPRIT) REI grant RR160093 (to S.G.E.). In addition, N.S. is a CPRIT Scholar in Cancer Research with funding from the Cancer Prevention and Research Institute of Texas (CPRIT) New Investigator Grant RR160021 and is supported by the Andrew Sabin Family Foundation Fellowship and NIH grant R35GM137836.

## Author contributions

This research project benefited from the collaborative efforts of the authors, with distinct contributions as follows. Conceptualization: .D.O., S.G.E. and M.L.K. Methodology: D.O. and M.L.K. Validation: D.O., A.C., U.G., A.S., T.M.P., C.H.L., W.A.M. and S.M.B. Resources: D.O., S.S.Y. and M.L.K. Data curation: D.O. Writing—original draft: D.O. Writing—review and editing: D.O., S.G.E. and M.L.K. Supervision: S.S.Y., N.S. and M.L.K. Project administration: S.S.Y. and M.L.K. Funding acquisition: D.O., A.C., S.G.E., T.M.P., C.H.L., W.A.M., S.S.Y. and M.L.K.

## Competing interests

D.O. is currently an employee of QIAGEN Digital Insights, QIAGEN, USA. The other authors declare no competing interests.

## Additional information

**Correspondence and requests for materials** should be addressed to Daniel Osorio, S. Stephen Yi or Marieke L. Kuijjer.

[1]Department of Oncology, Livestrong Cancer Institutes, Dell Medical School, The University of Texas at Austin, Austin, TX, USA. [2]Division of Medical Oncology, University of Colorado Cancer Center, School of Medicine, University of Colorado, Aurora, CO, USA. [3]Department of Immunology, Center for Systems Immunology, University of Pittsburg, Pittsburg, PA, USA. [4]Department of Epigenetics and Molecular Carcinogenesis, The University of Texas, MD Anderson Cancer Center, Houston, TX, USA. [5]Department of Bioinformatics and Computational Biology, The University of Texas, MD Anderson Cancer Center, Houston, TX, USA. [6]Interdisciplinary Life Sciences Graduate Programs (ILSGP), College of Natural Sciences, The University of Texas at Austin, Austin, TX, USA. [7]Oden Institute for Computational Engineering and Sciences (ICES), The University of Texas at Austin, Austin, TX, USA. [8]Department of Biomedical Engineering, Cockrell School of Engineering, The University of Texas at Austin, Austin, TX, USA. [9]Centre for Molecular Medicine Norway (NCMM), University of Oslo, Oslo, Norway. [10]Department of Pathology, Leiden University Medical Center (LUMC), Leiden University, Leiden, The Netherlands. [11]Leiden Center for Computational Oncology, Leiden University Medical Center (LUMC), Leiden University, Leiden, The Netherlands. ✉e-mail: daniecos@uio.no; stephen.yi@austin.utexas.edu; marieke.kuijjer@ncmm.uio.no

S. Stephen Yi
Marieke L. Kuijjer

# Reporting Summary

## Statistics

For all statistical analyses, confirm that the following items are present in the figure legend, table legend, main text, or Methods section.

| n/a | Confirmed | |
|---|---|---|
| ☐ | ☒ | The exact sample size (*n*) for each experimental group/condition, given as a discrete number and unit of measurement |
| ☐ | ☒ | A statement on whether measurements were taken from distinct samples or whether the same sample was measured repeatedly |
| ☐ | ☒ | The statistical test(s) used AND whether they are one- or two-sided *Only common tests should be described solely by name; describe more complex techniques in the Methods section.* |
| ☐ | ☒ | A description of all covariates tested |
| ☐ | ☒ | A description of any assumptions or corrections, such as tests of normality and adjustment for multiple comparisons |
| ☐ | ☒ | A full description of the statistical parameters including central tendency (e.g. means) or other basic estimates (e.g. regression coefficient) AND variation (e.g. standard deviation) or associated estimates of uncertainty (e.g. confidence intervals) |
| ☐ | ☒ | For null hypothesis testing, the test statistic (e.g. *F*, *t*, *r*) with confidence intervals, effect sizes, degrees of freedom and *P* value noted *Give P values as exact values whenever suitable.* |
| ☒ | ☐ | For Bayesian analysis, information on the choice of priors and Markov chain Monte Carlo settings |
| ☒ | ☐ | For hierarchical and complex designs, identification of the appropriate level for tests and full reporting of outcomes |
| ☐ | ☒ | Estimates of effect sizes (e.g. Cohen's *d*, Pearson's *r*), indicating how they were calculated |

*Our web collection on statistics for biologists contains articles on many of the points above.*

## Software and code

Policy information about availability of computer code

| Data collection | Lee et al. accessible through GEO: GSE132465, and GSE144735. Qi et al. accessible through GSA: HRA000979, Qian et al. accessible through ArrayExpress: E-MTAB-8107, and Che et al. accessible through GEO: GSE178318. Gene expression quantification of the patient-derived xenografts generated for this study are available as Supplementary Table 8. |
|---|---|
| Data analysis | R 4.1.2<br>Seurat 4.4.0<br>Nebulosa 1.12.0<br>Rfast 2.1.0<br>TABIX 0.2.6<br>STAR 2.7.11 |

For manuscripts utilizing custom algorithms or software that are central to the research but not yet described in published literature, software must be made available to editors and reviewers. We strongly encourage code deposition in a community repository (e.g. GitHub). See the Nature Portfolio guidelines for submitting code & software for further information.

## Data

Policy information about <u>availability of data</u>

All manuscripts must include a <u>data availability statement</u>. This statement should provide the following information, where applicable:

- Accession codes, unique identifiers, or web links for publicly available datasets
- A description of any restrictions on data availability
- For clinical datasets or third party data, please ensure that the statement adheres to our <u>policy</u>

> The following datasets were used to construct the colorectal cancer single-cell RNA-seq atlas used in this study: Lee et al. accessible through GEO: GSE132465, and GSE144735. Qi et al. accessible through GSA: HRA000979, Qian et al. accessible through ArrayExpress: E-MTAB-8107, and Che et al. accessible through GEO: GSE178318. Gene expression quantification of the patient-derived xenografts generated for this study are available as Supplementary Table 8. All the generated networks are available as independent files at https://github.com/dosorio/SCORPION/Results/Networks. We also made available the unrefined networks for human (hg38) and mice (mm10) genes through the the following link: https://github.com/dosorio/SCORPION/Data.

## Research involving human participants, their data, or biological material

Policy information about studies with <u>human participants or human data</u>. See also policy information about <u>sex, gender (identity/presentation), and sexual orientation</u> and <u>race, ethnicity and racism</u>.

| | |
|---|---|
| Reporting on sex and gender | N/A. This study does not involve human research participants. |
| Reporting on race, ethnicity, or other socially relevant groupings | N/A |
| Population characteristics | N/A |
| Recruitment | N/A |
| Ethics oversight | N/A |

Note that full information on the approval of the study protocol must also be provided in the manuscript.

# Field-specific reporting

Please select the one below that is the best fit for your research. If you are not sure, read the appropriate sections before making your selection.

☒ Life sciences    ☐ Behavioural & social sciences    ☐ Ecological, evolutionary & environmental sciences

For a reference copy of the document with all sections, see <u>nature.com/documents/nr-reporting-summary-flat.pdf</u>

# Life sciences study design

All studies must disclose on these points even when the disclosure is negative.

| | |
|---|---|
| Sample size | Various experiments were conducted using public data, with reported sample sizes specified for each experiment. The sample sizes were determined based on the availability of public data. |
| Data exclusions | Multiple quality control filters were performed across experiments. Excluded samples are reported in the manuscript. |
| Replication | Two independent cohorts were used to replicate the findings from single-cell RNA-seq data |
| Randomization | The manuscript provides descriptions of the covariates used to categorize samples. These were primarily employed for conducting differential analysis between regions and sides of the disease. |
| Blinding | The research was conducted without blinding. Groups were essential for performing regression analysis and supervised comparisons between meaningful groups. |

# Reporting for specific materials, systems and methods

We require information from authors about some types of materials, experimental systems and methods used in many studies. Here, indicate whether each material, system or method listed is relevant to your study. If you are not sure if a list item applies to your research, read the appropriate section before selecting a response.

## Materials & experimental systems

| n/a | Involved in the study |
|---|---|
| ☒ | ☐ Antibodies |
| ☒ | ☐ Eukaryotic cell lines |
| ☒ | ☐ Palaeontology and archaeology |
| ☐ | ☒ Animals and other organisms |
| ☒ | ☐ Clinical data |
| ☒ | ☐ Dual use research of concern |
| ☒ | ☐ Plants |

## Methods

| n/a | Involved in the study |
|---|---|
| ☒ | ☐ ChIP-seq |
| ☒ | ☐ Flow cytometry |
| ☒ | ☐ MRI-based neuroimaging |

# Animals and other research organisms

Policy information about [studies involving animals](); [ARRIVE guidelines]() recommended for reporting animal research, and [Sex and Gender in Research]()

| | |
|---|---|
| Laboratory animals | Mus musculus; Nu/Nu mice, 5 to 6 weeks old (Envigo). Mice had unrestricted access to food and water and were group-housed in a controlled environment with temperature (21-22°C), humidity (40-51%), and light (12/12 light/dark cycle) within the vivariums. |
| Wild animals | N/A; This study did not involve wild animals. |
| Reporting on sex | N/A; This study did not perform comparisons based on sex. |
| Field-collected samples | N/A; This study did not involve field-collected samples. |
| Ethics oversight | The University of Texas at Austin and The University of Colorado Institutional Animal Care and Use Committee approved all animal procedures. |

Note that full information on the approval of the study protocol must also be provided in the manuscript.

# Plants

| | |
|---|---|
| Seed stocks | *Report on the source of all seed stocks or other plant material used. If applicable, state the seed stock centre and catalogue number. If plant specimens were collected from the field, describe the collection location, date and sampling procedures.* |
| Novel plant genotypes | *Describe the methods by which all novel plant genotypes were produced. This includes those generated by transgenic approaches, gene editing, chemical/radiation-based mutagenesis and hybridization. For transgenic lines, describe the transformation method, the number of independent lines analyzed and the generation upon which experiments were performed. For gene-edited lines, describe the editor used, the endogenous sequence targeted for editing, the targeting guide RNA sequence (if applicable) and how the editor was applied.* |
| Authentication | *Describe any authentication procedures for each seed stock used or novel genotype generated. Describe any experiments used to assess the effect of a mutation and, where applicable, how potential secondary effects (e.g. second site T-DNA insertions, mosiacism, off-target gene editing) were examined.* |

