## [Peer Review File · Nature Computational Science]

Peer Review Information

Journal: Nature Computational Science

Manuscript Title: Population-level comparisons of gene regulatory networks modeled on high-throughput single-cell transcriptomics data

Corresponding author name(s): Daniel Osorio, Stephen Yi, Marieke Kuijjer

Editorial Notes: None

Reviewer Comments & Decisions:

Decision Letter, initial version:

Dear Dr Kuijjer,

Your manuscript "Population-level comparisons of gene regulatory networks modeled on high-throughput single-cell transcriptomics data" has now been seen by 2 referees, whose comments are appended below. Unfortunately, we were not able to get a 3rd referee on time, and we wanted to avoid delaying the decision on your paper. Please note that, at our discretion, we might try to get a 3rd opinion on the paper in future revisions.

You will see that, while the 2 referees find your work of interest, they have raised points that need to be addressed before we can make a decision on publication. The referees' reports seem to be quite clear. Naturally, we will need you to address **all** of the points raised. While we ask you to address all of the points raised, the following points need to be substantially worked on:

- Please better discuss the methodological details of the proposed approach and make sure that the manuscript is self-contained.
- Please better discuss the comparisons against related work: which aspects of SCORPION are likely to be responsible for improvements over alternatives and which aspects are maybe inferior? This will help editors and referees to better understand the methodological novelty and impact of the approach.

Please use the following link to submit your revised manuscript and a point-by-point response to the referees' comments (which should be in a separate document to any cover letter):

[Redacted]

** This url links to your confidential homepage and associated information about manuscripts you may have submitted or be reviewing for us. If you wish to forward this e-mail to co-authors, please delete this link to your homepage first. **

To aid in the review process, we would appreciate it if you could also provide a copy of your

manuscript files that indicates your revisions by making use of Track Changes or similar mark-up tools. Please also ensure that all correspondence is marked with your Nature Computational Science reference number in the subject line.

In addition, please make sure to upload a Word Document or LaTeX version of your text, to assist us in the editorial stage.

To improve transparency in authorship, we request that all authors identified as 'corresponding author' on published papers create and link their Open Researcher and Contributor Identifier (ORCID) with their account on the Manuscript Tracking System (MTS), prior to acceptance. ORCID helps the scientific community achieve unambiguous attribution of all scholarly contributions. You can create and link your ORCID from the home page of the MTS by clicking on 'Modify my Springer Nature account'. For more information please visit www.springernature.com/orcid.

We hope to receive your revised paper within three weeks. If you cannot send it within this time, please let us know.

Best,
Fernando

--

Fernando Chirigati, PhD
Chief Editor, Nature Computational Science
Nature Portfolio

Reviewers comments:

Reviewer #1 (Remarks to the Author):

In their manuscript "Population-level comparisons of gene regulatory networks modeled on high-throughput single-cell transcriptomics data" Osorio et al introduce a network inference framework SCORPION for the inference and analysis of transcriptional networks. They evaluate their approach in synthetic and curated data sets and they apply it to colorectal cancer data.

The manuscript is generally well written and the approach gives some useful insights into real data. As always with network inference methods the validation of an approach is to similar degree art as it is science.

The SCORPION algorithm is very briefly outlined; as much of it relies on PANDA, this may be excusable but I found it hard to follow and understand the approach, and its advantages over others. Fig 1B contains performance metrics for a suite of standard and not-so-standard approaches. Some of these alternatives require little information (PPCOR and PIDC) yet are apparently not far in

performance from SCORPION. Some of the more involved methods used in this comparison do much worse.

When reading the paper I was hoping for a better explanation which aspect of SCORPION is likely to be responsible for improvements over alternatives and which aspects is maybe inferior. The way PANDA constructs co-regulatory networks, for example, is pretty crude - in my view unacceptably crude - and I would expect that the STRING and motif footprints may give rise to the performance increases.

Without understanding these factors better I am unable to judge some of the downstream analysis. The authors have done a nice job of explaining what they did and how their results can be biologically interpreted but the overall uncertainty of the methodology always plays at the back of my mind.

Reviewer #2 (Remarks to the Author):

The authors introduce a tool, "scorpion", to construct gene regulatory networks from single cell/nuclei RNA-seq data. They use first synthetic data and then experimental data from an atlas to construct such networks. They compare the results to understand the difference of networks in different healthy and diseased tissues.

It is highly appreciated that they do not use averaged or bulk data, but focus on data from single cells. The authors integrate diverse type of data such as transcriptomics from single-cell-RNASeq, protein-protein interactions from the String database and regulatory interactions based on transcription factor binding motifs using data from ENSEMBL and TABIX databases. As a result, they obtain networks between the transcription factors and their targets. Moreover, the connections are weighted and specific for different data sets representing different cell types, treatments, sources or positions relative to a tumor.

Overall, their approach results in gene expression networks that can be compared. The authors also demonstrate that the results can be interpreted within the light of literature information, e.g. for different tumors like colorectal tumors, tumor positions, and so on.

I have a few minor points that I would like to see addressed, but in general find the work interesting and worth to publish.

Points

In their introduction, first sentence, the authors claim that transcription factors regulate expression of their targets in abundance-dependent manner. To my best knowledge, it is not only abundance, but also binding states and post-translational modifications that fine-tune activity of transcription factors. The interesting question resulting from this is how does this information enter the analysis, because it is not necessarily part of database information employed. Still the results are convincing. Thus, how is TF activity represented beyond RNA abundance?

Is initial unrefined network the same as original unrefined network?

Figure 2 Hnf4a/g \diamond Hnf4alpha/gamma, Fig2c and g: alpha and gamma seems to mixed up.

Page 3, right bottom paragraph states: "We found 221 and 211 large changes (out of the 95% confidence interval, 181 genes shared, Jaccard Index = 0.819) after perturbation in Hnf4a and Hnf4y outdegrees, respectively."

It is not explained how the outdegrees have been perturbed. Without this information, understanding of the figures is limited.

The information gathered on both healthy and tumor cells should be sufficient to not only compare different states, but also suggest reference networks, specifically for healthy cells. For many purposes, it would be very interesting to have such networks.

Since the result, before analysis and interpretation, of the network reconstruction is essentially a matrix showing connections between TFs and their target with a weight, it would also be useful to show /provide such a matrix to the reader.

How to make use of these tools? The Data Availability paragraph refers to the github repository stating "data and code required to replicate the analysis...". Is it straightforward not only to replicate, but to apply the tool to new data sets?

Can the tool also provide suggestions how to compare to protein abundance data, given that it is proteins that act in cells, not RNAs?

Minor

Explain your frequently used measures such as NES, FDR, FPKM, CPM

Author Rebuttal to Initial comments

Reviewer 1

In their manuscript "Population-level comparisons of gene regulatory networks modeled on high throughput single-cell transcriptomics data" Osorio *et al.*, introduce a network inference framework *SCORPION* for the inference and analysis of transcriptional networks. They evaluate their approach in synthetic and curated data sets and they apply it to colorectal cancer data. The manuscript is generally well written and the approach gives some useful insights into real data. As always with network inference methods the validation of an approach is to similar degree art as it is science. The *SCORPION* algorithm is very briefly outlined; as much of it relies on PANDA, this may be excusable but I found it hard to follow and understand the approach, and its advantages over others. Fig 1B contains performance metrics for a suite of standard and not-so-standard approaches. Some of these alternatives require little information (PPCOR and PIDC) yet are apparently not far in performance from *SCORPION*. Some of the more involved methods used in this comparison do much worse. When reading the paper I was hoping for a better explanation which aspect of *SCORPION* is likely to be responsible for improvements over alternatives and which aspects is maybe inferior. The way PANDA constructs co-regulatory networks, for example, is pretty crude - in my view unacceptably crude - and I would expect that the STRING and motif footprints may give rise to the performance increases. Without understanding these factors better I am unable to judge some of the downstream analysis. The authors have done a nice job of explaining what they did and how their results can be biologically interpreted but the overall uncertainty of the methodology always plays at the back of my mind.

1. The *SCORPION* algorithm is very briefly outlined.

Response: We greatly value the feedback provided by the reviewer.

Changes: Following the reviewer's comment, we have incorporated additional descriptions and mathematical formulas into the explanation of the *SCORPION* algorithm subsection of the results. The new section now reads as: *SCORPION* is an R package that generates through five iterative steps comparable, fully connected, weighted and directed transcriptome-wide gene regulatory networks from single-cell transcriptomic data that are suitable for their use in population-level studies (Figure 1A). To begin, the highly-sparse high-throughput single-cell/nuclei RNA-seq data is coarse-grained by collapsing a k number of more similar cells identified at the low dimensional representation of the multidimensional RNA-seq data. This approach reduces sample size while also decreasing data sparsity, allowing us better to capture the strength of the relationship between genes' expression.

The second step is to construct three distinct initial unrefined networks, as described in the PANDA algorithm: the co-regulatory network ($C^{(0)}$), the cooperative network ($P^{(0)}$), and the regulatory network ($W^{(0)}$). The co-regulatory network shows the co-expression patterns between genes. It is defined by Pearson correlation coefficients calculated for each pair of genes using the computed coarse-grained expression profiles. This network is constructed using correlation analyses over the generated coarse-grained transcriptomic data. The cooperative network accounts for the known protein-protein interactions between transcription factors. This information is downloaded from the STRING database. The third network is the unrefined regulatory network that describes the relationship between transcription factors and their target genes through transcription factors footprint motifs found in the promoter region of each gene.

After constructing the three unrefined networks, we employ a modified version of the Tanimoto similarity that allows to incorporate continuous values. This modified version is described by Equation 1, where x and y denote vectors of values that have been normalized to z-score units. The Tanimoto similarity is used to determine the agreement between the data represented by multiple networks.

$$T_Z = \frac{\sum_i x_i y_i}{\sqrt{\sum_i x_i^2 + \sum_i y_i^2 - |\sum_i x_i y_i|}} \quad (1)$$

We start by generating the availability network $A_{ij} = T_Z \left(W_i^{(t)}, C_j^{(t)} \right)$, representing the information flow from a transcription factor i to a gene j , using the accumulated evidence for how strongly the transcription factor influences the expression level of that gene ($W_j^{(t)}$), taking into account the behavior of other genes potentially targeted by that transcription factor ($C_j^{(t)}$). In addition, the responsibility network $R_{ij} = T_Z \left(P_i^{(t)}, W_j^{(t)} \right)$ is generated by computing the similarity between the cooperativity network and the regulatory network. The responsibility represents the information flowing from a transcription factor i to a gene j and captures the accumulated evidence for how strongly the gene j is influenced by the activity of that specific transcription factor ($W_j^{(t)}$), taking into account other potential regulators ($P_i^{(t)}$) of gene j .

The average ($\tilde{W}_{ij}^{(t)} = 0.5A_{ij}^{(t)} + 0.5R_{ij}^{(t)}$) of the availability and the responsibility networks is computed in the fourth step, and the regulatory network is updated ($\tilde{W}_{ij}^{(t+1)} = (1 - \alpha) W_{ij}^{(t)} + \alpha \tilde{W}_{ij}^{(t)}$) to include a user defined proportion ($\alpha = 0.1$ by default) of the information provided by the other two unrefined networks.

The cooperativity and co-regulatory networks are also updated in the fifth step using the new information contained in the updated regulatory network. Steps three through five are repeated iteratively (t) until the hamming distance between the networks reaches a user-defined threshold (0.001 by default). When convergence is reached, the refined regulatory network is returned as a matrix with transcription factors in the rows and target genes in the columns. The matrix values encode the strength of the relationship between each transcription factor and gene.

2. I found it hard to follow and understand the approach, and its advantages over others. Fig 1B contains performance metrics for a suite of standard and not-so-standard approaches. Some of these alternatives require little information (PPCOR and PIDC) yet are apparently not far in performance from *SCORPION*.
Response: We sincerely appreciate the valuable feedback provided by the reviewer. In order to evaluate the impact of data desparsification in *SCORPION* on downstream network modeling, we conducted a comprehensive benchmark analysis. Our objective was to compare *SCORPION*'s performance with that of other algorithms. To achieve this, we systematically evaluated various network construction algorithms by measuring their ability to recover gene-to-gene relationships, as represented by the co-regulatory matrix. For this purpose, we utilized BEELINE, a tool specifically designed for evaluating network construction methods. In our analysis, *SCORPION* was tested alongside 12 different algorithms. To ensure a fair comparison, we assessed each method's performance based on its ability to recover ground-truth interactions between genes. These ground-truth interactions were generated using default parameters, without incorporating any additional information apart from the expression matrix. Addressing the reviewer's comment, it is noted that while PPCOR and PIDC demonstrate comparable performance to *SCORPION*, they do not allow for the evaluation of all the regulatory mechanisms expected to be represented in a gene regulatory network. We hope this clarification provides a clearer understanding of our methodology for the BEELINE benchmark analysis and the comparison of *SCORPION* with other algorithms.

Changes: We added a statement in the *SCORPION* outperforms 12 other algorithms for single-cell gene regulatory network construction subsection. It now reads as: "Furthermore, in our analysis, we found that while PPCOR and PIDC exhibit similar performance to *SCORPION*, they are limited in their ability to evaluate all the regulatory mechanisms expected to be represented in a gene regulatory network. Additionally, when compared to other methods using seven different metrics related to network construction, *SCORPION* consistently ranks first on average".

3. When reading the paper I was hoping for a better explanation which aspect of *SCORPION* is likely to be responsible for improvements over alternatives and which aspects is maybe inferior.

Response: We greatly appreciate the valuable feedback provided by the reviewer. We understand the importance of providing a more comprehensive explanation in our paper regarding the specific aspects of *SCORPION* that contribute to its improvements over alternative methods, as well as any potential limitations or areas where it may be inferior. To highlight the factors that likely contribute to the improvements in *SCORPION*, we would like to emphasize three key aspects. Firstly, *SCORPION* effectively addresses sparsity in single-cell gene expression matrices by utilizing a coarse-graining technique. This approach enhances the detection of the underlying correlation structure within the gene regulatory network, leading to improved performance. Secondly, *SCORPION* allows for the construction of comparable gene regulatory networks from single-cell/nuclei RNA-seq data, enabling population-level comparisons. This is achieved by leveraging the same baseline priors, ensuring consistency and facilitating meaningful analyses across different datasets. Thirdly, *SCORPION* offers notable computational improvements compared to other gene regulatory network construction tools. By default, it utilizes sparse matrices, resulting in reduced memory usage and faster matrix multiplications. Additionally, it incorporates truncated principal components for the desparsification step, further enhancing computational efficiency. Finally, *SCORPION* is readily available on multiple platforms through the CRAN repositories, simplifying its installation on various operating systems. It is of course also important to acknowledge certain factors that may make *SCORPION* less favorable compared to other tools. Specifically, our method requires a priori information, which is currently only available for well-studied model organisms. Consequently, the applicability of *SCORPION* is limited to cases where protein-protein interaction (PPI) and transcription factor (TF) motif scan data exist. These clarifications aim to provide a more comprehensive understanding of *SCORPION*'s strengths, computational improvements, and limitations, as well as the specific considerations that need to be taken into account when applying the method.

Changes: We modified the last two paragraphs of the methodology that now read as: *SCORPION* also offers notable computational improvements compared to other gene regulatory network construction tools. By default, it utilizes sparse matrices, resulting in reduced memory usage and faster matrix multiplications. Additionally, it incorporates truncated principal components for the desparsification step, further enhancing computational efficiency. Furthermore, *SCORPION* is readily available on multiple platforms through the CRAN repositories, simplifying its installation and use on various operating systems.

Finally, *SCORPION* enables the use of the same statistical techniques that account for population heterogeneity and are widely used in other areas of genomics data analysis by constructing very precise and highly comparable gene regulatory networks for each sample. We anticipate that *SCORPION* will be used not only to characterize molecular mechanisms driving phenotypes, but also to investigate a wide range of important questions in precision medicine, health, and biomedical research now that gene regulatory network perturbations have been shown to be effective at reproducing experimental results.

4. The way PANDA constructs co-regulatory networks, for example, is pretty crude - in my view unacceptably crude - and I would expect that the STRING and motif footprints may give rise to the performance increases. **Response:** We appreciate the reviewer for bringing attention to the insufficient level of detail in the manuscript regarding the construction of the co-regulatory networks. We would also like to clarify to the reviewer that, as described in the manuscript, after the three initial networks are constructed, additionally to the regulatory network, *The cooperativity and co-regulatory networks are also updated in the fifth step using the new information contained in the updated regulatory network.* This procedure is repeated until convergence, and allows to the initially constructed co-regulatory network to be refined by incorporating the information provided by the protein-protein interaction networks and the promoter associated motif information during the optimization procedure. Since integrating multiple sources of information has shown to provide robust approximation and high performance of gene regulatory network structure inference across diverse datasets, as described in *Daniel Marbach, James C Costello, Robert Küffner, Nicole M Vega, Robert J Prill, Diogo M Camacho, Kyle R Allison, Manolis Kellis, James J Collins, and Gustavo Stolovitzky. Wisdom of crowds for robust gene network inference. Nature methods, 9(8):796-804, 2012.*, the refined co-regulatory network returned by *SCORPION* accounts for the agreement among the three data types used as input.

Changes: Following the reviewer's suggestion we added more information to the methodology used by *SCORPION* to construct the co-regulatory network. The sentence now reads as: "The co-regulatory network shows the co-expression patterns between genes. It is defined by Pearson correlation coefficients calculated for each pair of genes using the computed coarse-grained expression profiles."

Reviewer 2

The authors introduce a tool, "*SCORPION*", to construct gene regulatory networks from single cell/nuclei RNA-seq data. They use first synthetic data and then experimental data from an atlas to construct such networks. They compare the results to understand the difference of networks in different healthy and diseased tissues. It is highly appreciated that they do not use averaged or bulk data, but focus on data from single cells. The authors integrate diverse type of data such as transcriptomics from single-cell-RNASeq, protein-protein interactions from the STRING database and regulatory interactions based on transcription factor binding motifs using data from ENSEMBL and TABIX databases. As a result, they obtain networks between the transcription factors and their targets. Moreover, the connections are weighted and specific for different data sets representing different cell types, treatments, sources or positions relative to a tumor. Overall, their approach results in gene expression networks that can be compared. The authors also demonstrate that the results can be interpreted within the light of literature information, e.g. for different tumors like colorectal tumors, tumor positions, and so on. I have a few minor points that I would like to see addressed, but in general find the work interesting and worth to publish.

Major Points

1. In their introduction, first sentence, the authors claim that transcription factors regulate expression of their targets in abundance-dependent manner. To my best knowledge, it is not only abundance, but also binding states and post-translational modifications that fine-tune activity of transcription factors.

Response: We fully agree with the reviewer. While the abundance of transcription factors can play a role in regulating gene expression, it is not the sole determining factor. In fact, this is something we previously showed in Sonawane, Abhijeet Rajendra, et al. "Understanding tissue-specific gene regulation." *Cell reports* 21.4 (2017): 1077-1088.

Changes: We have modified the first sentence of the introduction to: In eukaryotes, gene expression is carefully regulated by transcription factors—proteins that play a crucial role in determining cell identity and controlling cellular states. They achieve this by either activating or repressing the expression of specific target genes. This regulation is dependent on the abundance of transcription factors, their ability to bind to chromatin (DNA-protein complex), and various post-translational modifications they undergo.

2. The interesting question resulting from this is how does this information enter the analysis, because it is not necessarily part of database information employed. Still the results are convincing. Thus, how is TF activity represented beyond RNA abundance?

Response: We appreciate the reviewer's comment, which brings attention to an untested feature of our tool. *SCORPION* incorporates three types of input data: gene expression data, prior networks for protein-protein interactions, and the relationship between transcription factors and their target genes through transcription factors footprint motifs. To refine this information, we utilize the coexpression matrix, which is derived from the expression matrix. With this data, we construct the availability network (A_{ij}), which captures the flow of information from a transcription factor i to a gene j . This network provides insights into the extent to which the transcription factor influences the expression level of the gene, considering the behavior of other genes that may also be targeted by the same transcription factor. It is worth noting that modifications to the binding potential of the transcription factors can also be modeled by altering the prior network as input to our approach.

Changes: We added the following sentence to the discussion section: The use of data other than gene expression distinguishes *SCORPION* from most other methodologies and allow for the modeling of known perturbations of protein-protein interactions and transcription-factors binding patterns.

3. Is initial unrefined network the same as original unrefined network?

Response: We thank the reviewer for pointing out a confusing terminology in the manuscript. We have removed the term 'original' from the problematic sentence.

Changes: The sentence now reads as: The average of the availability and the responsibility networks is computed in the fourth step, and the regulatory network is updated to include a user defined proportion ($\alpha = 0.1$ by default) of the information provided by the other two unrefined networks.

4. Figure 2 Hnf4a/g and Hnf4alpha/gamma, Fig 2C and 2G: alpha and gamma seems to mixed up.

Response: We appreciate the reviewer's feedback regarding a confusing terminology in the manuscript. In Figure 2C and 2G, the axis labels are meant to represent the genotype from the experimental work of Chen *et al.*, (2019) associated with the computed network. Consequently, the legend of Figure 2C is described as follows: Spearman correlation ($\hat{\rho}$) of the outdegrees for the *Hnf4 α* transcription factor in *Hnf4 α γ ^{WT}* and *Hnf4 α γ ^{DKO}* mice intestinal epithelium cells. Genes falling outside the 95% confidence interval are color-coded and labeled accordingly (in red for up-regulated and in blue for down-regulated). The legend of Figure 2G is described as follows: Spearman correlation ($\hat{\rho}$) of the outdegrees for the *Hnf4 γ* transcription factor in *Hnf4 α γ ^{WT}* and *Hnf4 α γ ^{DKO}* mice intestinal epithelium cells. Genes out of the 95% confidence interval are color coded and labeled (in red if up-regulated, and in blue if down-regulated).

5. Page 3, right bottom paragraph states: "We found 221 and 211 large changes (out of the 95% confidence interval, 181 genes shared, Jaccard Index = 0.819) after perturbation in Hnf4 α and Hnf4 γ outdegrees, respectively." It is not explained how the outdegrees have been perturbed. Without this information, understanding of the figures is limited.

Response: We thank the reviewer for pointing out a confusing terminology in the manuscript. Here, we refer to experimental perturbation.

Changes: We modified the problematic sentence to add more details about the origin of the perturbation and now the sentence reads as: "We identified 221 and 211 large changes (out of the 95% confidence interval, 181 genes shared, Jaccard Index = 0.819) after the experimental perturbation of *Hnf4 α* and *Hnf4 γ* , respectively."

6. The information gathered on both healthy and tumor cells should be sufficient to not only compare different states, but also suggest reference networks, specifically for healthy cells. For many purposes, it would be very interesting to have such networks.

Response: We agree with the reviewer, for that reason, we made all the 650 generated networks for the colon-rectal cancer analyses accessible as R objects available using the following link: Link to Networks.

7. Since the result, before analysis and interpretation, of the network reconstruction is essentially a matrix showing connections between TFs and their target with a weight, it would also be useful to show/provide such a matrix to the reader.

Response: We agree with the reviewer, for that reason, we made all the generated networks for this project accessible as R objects available using the following link: Link to R Objects. We also made available the unrefined networks for human (hg38) and mice (mm10) genes through the the following link: Link to Networks.

8. How to make use of these tools? The Data Availability paragraph refers to the github repository stating "data and code required to replicate the analysis...". Is it straightforward not only to replicate, but to apply the tool to new data sets?

Response: We appreciate the reviewer's interest in applying *SCORPION* to new datasets. We have made the package available at CRAN for multiplatform install using: Link to CRAN, and also as a repository <https://github.com/kuijjerlab/SCORPION> with documented functions and examples that guide the user to apply the method to new datasets.

9. Can the tool also provide suggestions how to compare to protein abundance data, given that it is proteins that act in cells, not RNAs?

Response: We thank the reviewer for raising an important point. *SCORPION* primarily focuses on analyzing high-throughput single-cell transcriptomics data, it does not directly provide suggestions on comparing to protein abundance data. As you correctly mentioned, proteins are the functional actors in cells, and their abundance can differ from RNA levels. Therefore, incorporating protein abundance data into our tool's analysis is currently outside its scope. However, we acknowledge the significance of this aspect and will consider it for future enhancements or potential collaborations with other tools that specifically address protein-level analysis.

Minor Points

1. Explain your frequently used measures such as NES, FDR, FPKM, CPM.

Response: We thank the reviewer for pointing out unexplained terminology in the manuscript.

Changes: In Figure 6F and 6G, we provided the following statement for clarity: "Expression levels are reported as fragments per kilobase of transcript per million mapped reads (FPKM) and counts per million (CPM), respectively." For Figure 4C and 4D, we added a statement explaining that the data was ranked based on the normalized enrichment score (NES). Furthermore, in the section on "Modeling the regulatory differences that drive colorectal cancer progression," we included information about our approach to addressing multiple testing. Specifically, we adjusted the P-values using the False Discovery Rate (FDR) method.

Decision Letter, first revision:

Dear Dr Kuijjer,

First of all, apologies for the long delay here. As you know, your manuscript "Population-level comparisons of gene regulatory networks modeled on high-throughput single-cell transcriptomics data" had been seen by 2 referees originally (Referees #1 and #2), and both referees were unavailable this time to comment on the revision. Therefore, we had to find 2 new referees (Referees #3 and #4), whose comments are appended below.

Both referees mentioned (privately) to our editorial team that the comments from Referees #1 and #2 were appropriately addressed. They also provided new comments that we would like to see addressed.

While we ask you to address all of the points raised, the following points need to be substantially worked on:

- Please better clarify the advantages of SCORPION when compared to the original PANDA framework.
- According to Referee #3, the population-level comparison aspect should be explained more clearly.
- Please show with experiments that Scorpion leverages the prior information effectively, as recommended by Referee #3.
- Please make sure to upload the supplementary tables, which are missing from the submission.
- Please better motivate the framework, as recommended by Referee #4.

Please use the following link to submit your revised manuscript and a point-by-point response to the referees' comments (which should be in a separate document to any cover letter):

[Redacted]

** This url links to your confidential homepage and associated information about manuscripts you may have submitted or be reviewing for us. If you wish to forward this e-mail to co-authors, please delete this link to your homepage first. **

To aid in the review process, we would appreciate it if you could also provide a copy of your manuscript files that indicates your revisions by making use of Track Changes or similar mark-up tools. Please also ensure that all correspondence is marked with your Nature Computational Science reference number in the subject line.

In addition, please make sure to upload a Word Document or LaTeX version of your text, to assist us in the editorial stage.

To improve transparency in authorship, we request that all authors identified as 'corresponding author' on published papers create and link their Open Researcher and Contributor Identifier (ORCID) with their account on the Manuscript Tracking System (MTS), prior to acceptance. ORCID helps the scientific community achieve unambiguous attribution of all scholarly contributions. You can create and link your ORCID from the home page of the MTS by clicking on 'Modify my Springer Nature account'. For more information please visit www.springernature.com/orcid.

We hope to receive your revised paper within three weeks. If you cannot send it within this time, please let us know.

Best,
Fernando

--

Fernando Chirigati, PhD
Chief Editor, Nature Computational Science
Nature Portfolio

Reviewers comments:

Reviewer #3 (Remarks to the Author):

In their manuscript "Population-level comparisons of gene regulatory networks modeled on high-throughput single-cell transcriptomics data", Osorio et al. present a new R package and method SCORPION, which is based on the PANDA method for reconstructing gene-regulatory networks using prior information on protein-protein interaction and TF-gene associations. PANDA uses a continuous form of the Tanimoto similarity coefficient to compute and optimize the similarities of three different types of networks, thus refining the baseline network information and revealing condition-specific gene-regulatory network changes. This approach makes elegant use of prior information during network construction, as opposed to other network construction methods which consider such information only for interpretation. As far as I can see, SCORPION extends PANDA by one more step before applying it to single-cell RNA-seq data. Namely, k similar cells are aggregated into SuperCells to overcome the sparse nature of RNA-seq data. The paper is overall well written and offers great application cases that are explored in depth.

Major:

- It is unclear if SCORPION has any advantages other than the aggregation into supercells compared to the original PANDA framework. Differences (also convenience features that might exist such as integration with Seurat, etc.) to the original PANDA method should be described in more detail. Some possible advantages are briefly mentioned in the discussion of the revised manuscript but these are in relation to other GRN methods and it is not clear how this compares to PANDA.
- The aggregation of sparse single-cell data is a common practice in many single-cell RNA-seq methods and is done under different names. It would help the readers to highlight that this procedure is also called "meta-cells" or (mini) pseudo-bulks across different publications. If the authors do something different than just aggregating the counts of the individual cells, this needs to be explained.
- The aggregation into Supercells considers k similar cells. It is not clear which k was used here and, since this is a hyperparameter, if the authors have tried different k s to achieve optimal results in the Beeline benchmark.
- The aggregation is likely beneficial for other methods for GRN construction. Since PPCOR already achieves similar performance, one wonders if feeding SuperCells into PPCOR would close the

remaining performance gap.

- The individual metrics of Beeline should be described in more detail, especially the two types of feed loops tested. Also, the mutual interactions are not understandable without reading the original Beeline paper. Please provide more details about these metrics.
- The authors mention in the title and in the discussion that SCORPION "enables the use of the same statistical techniques that account for population heterogeneity and are widely used in other areas of genomics data analysis". I feel that it remains too vague what techniques the authors refer to here. The population-level comparison aspect should be explained more clearly. For me it currently leads to confusion as to what exactly is meant here.
- The t-test is frequently used in the manuscript. I think this may be acceptable for comparing the weights but for differential expression analysis, this does not seem a good choice for me as there are better metrics for differential expression analysis in single-cell RNA-seq data.
- It would be great if the authors could show that Scorpion leverages the prior information effectively. This could be achieved by comparing Scorpion on the real input compared to randomized baselines, e.g. shuffled TF-gene and protein-protein interactions.
- I could not find the supplemental tables in the submission system, they were also not in the preprint.

Minor:

- Typo: transcri*p*tion factor (page 10)
- The sentence in the discussion "Despite the fact..." does not make sense overall. Please rephrase and restructure.
- Introduction: "Depending on the level of detail of the transcriptomic data" / not clear what you mean here by level of detail.
- "an aggregate network is built using all of the transcriptomes from each experimental group to represent each one of them" / also not clear what is meant here, please rephrase.
- The sentence "This network is constructed using correlation analyses over the generated coarse-grained transcriptomic data" represents a duplication in the revised manuscript.
- Tanimoto is not followed by the word similarity anymore in the revised manuscript.

Reviewer #3 (Remarks on code availability):

The code is on CRAN and can be easily installed from there. Additional usage information is rather sparse, e.g. a vignette demonstrating the use would be nice. At the very least, the README on the github repo should guide the user more towards using the code in the repository.

Reviewer #4 (Remarks to the Author):

The authors present SCORPION - a new method to reconstruct "gene regulatory networks" from single cell RNASeq data.

The paper describes the method, some results on simulated data, showing superiority over other tools and then some potential usage of the tool on different real scRNASeq data.

As for the method - it is a relatively complex iterative approach that begins by constructing 3 different networks: cooperativity net., regulatory net., and co-regulatory network. Then it iteratively uses a "tanimoto similarity inspired" formula to iteratively "refine these networks until convergence.

I think it is potentially interesting work that might be a step forward for the community of network reconstruction, however, there is a number of issues that need to be resolved before the paper is ready for publication, in my opinion.

The benchmarking against other tools shows quite impressive numbers, in comparison with other tools, but not so impressive overall. The AUROC is close to 0.6 for all of the methods (ranging between 0.62 to 0.55, where 0.5 is essentially random). This is not to criticize the work - it seems to be a step forward - but the authors should consider toning down some of the expressions they use to describe their results. In particular the mentions of "breakthrough in gene regulatory network construction" are a bit over-the-top.

The methodology is relatively well described in terms of what is done, but is not so well described in terms of why is it done the way it is. The whole T_Z function is a bit mysterious for me. It seems to be some sort of matrix multiplication with normalization built-in. I do not understand the motivation for using it to generate both the R and A matrices or the reason that they are then averaged to give the delta to the new iteration of the W matrix. Also, the notion that the C and P networks are also updated is unclear. Without the understanding of what is the reason to this particular approach, the whole work seems arbitrary and even if the results are quite impressive, it is not really a step forward for the field.

The part regarding the HNF factors changes upon DKO experiment are not really very convincing to me. Indeed the authors observe a shift downwards in the distribution of out-degrees of the genes knocked out. On one hand - there is a significant difference in the direction that we expect. On the other hand - these genes are knocked out in the actual data. Why do they have any out degrees in these data at all? (and BTW. I do not understand what are these distributions over. I would expect an outdegree, defined as a sum of all weight out of a single node to be a number, not a distribution).

The third part, concerned with the colorectal cancer data is quite interesting. The authors show that they are quite creative in their use of the results of the SCORPION method. And some of the results they show (like the differential network analysis or regulatory differences between left- and right-sided cancer cells are very interesting and quite a creative way to use the tool they have.

However, as this is a methodological paper, I do not think that the method is described well enough. The use of the T_Z function and the whole process needs to be explained better, and - even more importantly - motivated for the paper to be a proper step forward in the field.

Author Rebuttal, first revision:

Reviewer 3

In their manuscript “Population-level comparisons of gene regulatory networks modeled on high-throughput single-cell transcriptomics data”, Osorio *et al* present a new R package and method *SCORPION*, which is based on the PANDA method for reconstructing gene-regulatory networks using prior information on protein-protein interaction and TF-gene associations. PANDA uses a continuous form of the Tanimoto similarity coefficient to compute and optimize the similarities of three different types of networks, thus refining the baseline network information and revealing condition-specific gene-regulatory network changes. This approach makes elegant use of prior information during network construction, as opposed to other network construction methods which consider such information only for interpretation. As far as I can see, *SCORPION* extends PANDA by one more step before applying it to single-cell RNA-seq data. Namely, k similar cells are aggregated into SuperCells to overcome the sparse nature of RNA-seq data. The paper is overall well written and offers great application cases that are explored in depth.

Major:

1. It is unclear if *SCORPION* has any advantages other than the aggregation into supercells compared to the original PANDA framework. Differences (also convenience features that might exist such as integration with Seurat, etc.) to the original PANDA method should be described in more detail. Some possible advantages are briefly mentioned in the discussion of the revised manuscript but these are in relation to other GRN methods and it is not clear how this compares to PANDA.

Response: We appreciate the reviewer’s comment. Additional to collapsing cells into Super/Meta-Cells to reduce data sparsity and allow for better correlation estimates, *SCORPION* offers notable computational improvements compared to other gene regulatory network construction tools, including the original PANDA algorithm. By default, it utilizes sparse matrices, resulting in reduced memory usage and faster matrix multiplications. Additionally, it incorporates truncated principal components for the desparsification step, further enhancing computational efficiency.

Changes: We modified the introduction and the discussion sections to better highlight the differences between *SCORPION*, PANDA and other methods for gene regulatory construction. Please refer to the document with tracked changes.

2. The aggregation of sparse single-cell data is a common practice in many single-cell RNA-seq methods and is done under different names. It would help the readers to highlight that this procedure is also called “meta-cells” or (mini) pseudo-bulks across different publications. If the authors do something different than just aggregating the counts of the individual cells, this needs to be explained.

Response: We agree with the reviewer’s comment, the aggregation of sparse single-cell data is a common practice in many single-cell RNA-seq methods and is done under different names. We initially chose to refer to this as SuperCells, as this was the term originally used in the BioRxiv publication by Bilous *et al*, which explains the method *SCORPION* incorporates. However, we agree with the reviewer that this could be described more clearly.

Changes: We have included the following sentence in the manuscript: “This procedure is sometimes also referred to as meta-cells or (mini) pseudo-bulks.” In addition, we replaced the instances where SuperCells are described to “Super/Meta-Cells” and added a reference to the original publication describing MetaCells (Bilous *et al*. 2022).

3. The aggregation into Supercells considers k similar cells. It is not clear which k was used here and, since this is a hyper-parameter, if the authors have tried different k s to achieve optimal results in the BEELINE benchmark.

Response: We appreciate the reviewer’s interest in our methodology. During the entire manuscript analyses we did not modify the default parameters at any point. The default parameters for constructing SuperCells are available at: <https://github.com/kuijjerlab/SCORPION/blob/main/R/makeSuperCells.R>

Changes: We have modified the methods section to include an statement disclosing the use of default parameters for each network generation approach.

4. The aggregation is likely beneficial for other methods for GRN construction. Since PPCOR already achieves similar performance, one wonders if feeding SuperCells into PPCOR would close the remaining performance gap.

Response: We thank the reviewer for their question. First, we would like to emphasize that our primary objective was to elucidate the implementation of *SCORPION* and its application to single-cell RNA-seq data, rather than evaluating the impact of cell aggregation on constructing gene regulatory networks.

However, in response to the reviewer’s inquiry, we used the same coarse-graining approach to the single-cell RNA-seq data employed in *SCORPION* in an application of the PPCOR method for the *Hnf4a* γ double knockout dataset. This was done to assess PPCOR’s performance under real-world scenarios. The results underscore the issue that motivated the development of *SCORPION*: the network modeled on the knock

out data minimally correlates with network modeled on the wild type data ($\hat{\rho} = 0.00096$, $P = 0.67$), and differential edge weights are normally distributed around 0. Upon scrutinizing the edge weights of the *Hnf4 α* transcription factor, we observed a few subtle differences. However, on average, these are not significantly distinct from 0 (as can be seen in the figure below, top row, third panel). For the *Hnf4 γ* transcription factor, the differences are, on average, slightly negative ($\hat{\rho} = -0.0048$, $P = 0.024$) and significantly different from 0. The correlation between edge weights in both (*Hnf4 α* γ^{WT} and *Hnf4 α* γ^{DKO}) networks for both transcription factors is very close to 0. Crucially, the enrichment of enterocyte markers that we obtained with *SCORPION* is not evident when using PPCOR ($P_{\text{adj}} = 0.92$ and 0.96). We have incorporated these results in the supplemental material of the manuscript.

Fig. 1: Evaluation PPCOR's performance in transcriptome-wide scenarios using the same coarse-graining approach implemented in *SCORPION*. The first row presents results for the *Hnf4 α* , and the second row for the *Hnf4 γ* transcription factor. In the first panel, the distribution of edge weights is displayed, while the second panel shows the distribution of paired differences in edge weights between the networks constructed for the *Hnf4 α* γ^{DKO} and *Hnf4 α* γ^{WT} samples. The third panel illustrates the correlation between edge weights in both networks, and the fourth panel showcases the results of gene set enrichment analysis using the paired differences in edge weights between the *Hnf4 α* γ^{DKO} and *Hnf4 α* γ^{WT} sample networks.

- The individual metrics of BEELINE should be described in more detail, especially the two types of feed loops tested. Also, the mutual interactions are not understandable without reading the original Beeline paper. Please provide more details about these metrics.

Response: We thank the reviewer for their comment and agree it would help the reader to have more detail related to these methods. For this reason, we have now added additional descriptions of the FBL, FFL and MI motif structures to the manuscript.

Changes: The legend of Figure 1B, and methods section related with the benchmarking using synthetic data, now reads as follows: "We compared algorithms based on their average performance across seven different metrics: AUROC (area under the receiver operating characteristic), AUPRC (area under the precision-recall curve), computing time, level bias due to expression level, FBL (feedback loops, where some portion (or all) of a regulatory response is used as input for future gene regulation), FFL (feed-forward loop, a three-gene pattern composed of two input transcription factors, one of which regulates the other, both of which jointly regulate a target gene), and MI (mutual iterations, equally weighted interactions between regulator-target and vice-versa) motif structure identification."

- The authors mention in the title and in the discussion that *SCORPION* "enables the use of the same statistical techniques that account for population heterogeneity and are widely used in other areas of genomics data analysis". I feel that it remains too vague what techniques the authors refer to here. The population-level comparison aspect should be explained more clearly. For me it currently leads to confusion as to what exactly is meant here.

Response: We appreciate the reviewer's comment. We included a definition of population-level analysis in the introduction of the manuscript. In addition, we added more details about the methodologies that are applicable for the analysis of the networks generated by *SCORPION*.

Changes: We added an explanation to "...suitable for population-level studies" in the introduction to clarify what we mean by population-level analyses: "...suitable for statistical analyses that leverage multiple

samples per experimental group—something we refer to in the remainder of this manuscript as “population-level studies.”” The discussion section now reads as: “Finally, by constructing precise and highly comparable gene regulatory networks for each sample, *SCORPION* enables the use of the same statistical techniques that consider population heterogeneity and are widely used in other areas of genomic data analysis. These methods include, but are not limited to, clustering based on sample similarity, dimensionality reduction, and differential analysis.”

7. The *t*-test is frequently used in the manuscript. I think this may be acceptable for comparing the weights but for differential expression analysis, this does not seem a good choice for me as there are better metrics for differential expression analysis in single-cell RNA-seq data.

Response: We appreciate the reviewer’s concern and wish to clarify that our analysis of gene expression levels served the purpose of cross-validating the results obtained from differential network analyses, to which methods developed for the analysis of single-cell transcriptomic data are not applicable, due to the different structure of the data. Thus, to be able to compare analyses with the same test, we chose the *t*-test, which is applicable to both.

8. It would be great if the authors could show that *SCORPION* leverages the prior information effectively. This could be achieved by comparing *SCORPION* on the real input compared to randomized baselines, e.g. shuffled TF-gene and protein-protein interactions.

Response: We greatly appreciate the reviewer’s feedback, as it is instrumental in addressing a previously unexamined aspect of the *SCORPION* gene regulatory network construction—the performance of data integration. Following the reviewer’s advice, we conducted a comprehensive assessment by introducing randomization into the prior data 50 times, each with different seeds. This allowed us to examine the impact of stochastic priors on network construction using the double knockout dataset. Our analysis revealed a pattern: the incorporation of random priors significantly reduced the disparities between the network representing the perturbed sample and the wild-type reference. This was evident both in the change in Spearman correlation coefficient and in the associated *p*-values. As expected, the Spearman correlation coefficient increased from 0.88 when using the original prior to an average of 0.95 with randomized ones, indicating that fewer perturbations were measured when using randomized priors. This difference was statistically significant (one-sided *t*-test $P = 2.2 \times 10^{-16}$). Additionally, there was a smaller average difference in the weights of both *Hnf4 α* (from -0.24 to -0.17 in average; one-sided *t*-test $P = 3.25 \times 10^{-11}$) and *Hnf4 γ* (from -0.21 to -0.17 in average; one-sided *t*-test $P = 2.51 \times 10^{-15}$) transcription factors with their target genes in networks using randomized priors. The figure displayed below shows the distribution of the values across the 50 randomizations with the value obtained using the original regulatory prior in red.

Fig. 2: Differences in results after employing random priors during the construction of single-cell gene regulatory networks with *SCORPION*. The presented density plots illustrate the distribution of the correlation ($\hat{\rho}$) between the edges’ weight correlation for both *Hnf4 α* and *Hnf4 γ* transcription factors. Additionally, the distribution of the average pair differences between the edges’ weight for *Hnf4 α* : $\hat{\mu}(\text{KO}_{Hnf4\alpha} - \text{WT}_{Hnf4\alpha})$, and *Hnf4 γ* : $\hat{\mu}(\text{KO}_{Hnf4\gamma} - \text{WT}_{Hnf4\gamma})$ transcription factors in networks constructed for the *Hnf4 α* ^{DKO} and *Hnf4 α* ^{WT} samples is shown. All distributions are based on 50 runs with random priors, with values obtained using correct priors highlighted in red.

Changes: We added the analysis described above, as well as the figure, within the subsection *SCORPION* accurately detects changes in transcription factor activity and their impact on target genes of the results.

9. I could not find the supplemental tables in the submission system, they were also not in the preprint.

Response: We thank the reviewer for pointing this out.

Changes: Additionally to the files shared through the submission system, all the results are now accessible for review through: <https://github.com/dosorio/SCORPION/>

Minor:

1. Typo: transcri*p*tion factor (page 10)

Response: We thank the reviewer for pointing this out.

Changes: We have fixed the typo.

- The sentence in the discussion "Despite the fact..." does not make sense overall. Please rephrase and restructure.

Response: We appreciate the reviewer's suggestion. We have modified the sentence to make it more readable.

Changes: The sentence now reads as follows: "While there are numerous methods available for building gene regulatory networks from single-cell transcriptomes, they often fall short in allowing comparative analysis to evaluate sample-to-sample variability in the regulatory mechanisms that define a phenotype at the population level."

- Introduction: "Depending on the level of detail of the transcriptomic data" / not clear what you mean here by level of detail.

Response: We appreciate the reviewer's suggestion. We have modified the sentence to make it more readable.

Changes: The sentence now reads as: "Depending on the set of cells or samples with transcriptomic data included in the gene regulatory network reconstruction, networks can either represent the regulatory programs of specific cell types within a tissue, or capture average mechanisms that define the entire tissue from which the sample was taken."

- "an aggregate network is built using all of the transcriptomes from each experimental group to represent each one of them" / also not clear what is meant here, please rephrase.

Response: We appreciate the reviewer's suggestion. We have modified the sentence to make it more readable.

Changes: The sentence now reads as: "In the context of differential network analysis, an aggregate network is often constructed by combining the transcriptomes of all cells within each experimental group. This network then represents the characteristics of each experimental group, and these aggregate network models can be used for comparative analysis."

- The sentence "This network is constructed using correlation analyses over the generated coarse-grained transcriptomic data" represents a duplication in the revised manuscript.

Response: We thank the reviewer for pointing this out.

Changes: We have removed the sentence from the manuscript.

- Tanimoto is not followed by the word similarity anymore in the revised manuscript.

Response: We thank the reviewer for pointing this out.

Changes: We have added the word similarity after Tanimoto.

Remarks on code availability:

- The code is on CRAN and can be easily installed from there. Additional usage information is rather sparse, e.g. a vignette demonstrating the use would be nice. At the very least, the README on the github repo should guide the user more towards using the code in the repository.

Response: We are grateful for the reviewer's suggestion. To address the reviewer's concern, we have now included a comprehensive example using a demo dataset in the README.md file within the repository containing the tool's source code. This example can be accessed using the following link:

<https://github.com/kuijjerlab/SCORPION/>

Reviewer 4

The authors present *SCORPION* - a new method to reconstruct "gene regulatory networks" from single cell RNA-seq data. The paper describes the method, some results on simulated data, showing superiority over other tools and then some potential usage of the tool on different real scRNASeq data. As for the method - it is a relatively complex iterative approach that begins by constructing 3 different networks: cooperativity net., regulatory net., and co-regulatory network. Then it iteratively uses a "tanimoto similarity inspired" formula to iteratively "refine these networks until convergence. I think it is potentially interesting work that might be a step forward for the community of network reconstruction, however, there is a number of issues that need to be resolved before the paper is ready for publication, in my opinion.

- The benchmarking against other tools shows quite impressive numbers, in comparison with other tools, but not so impressive overall. The AUROC is close to 0.6 for all of the methods (ranging between 0.62 to 0.55, where 0.5 is essentially random). This is not to criticize the work - it seems to be a step forward - but the authors should consider toning down some of the expressions they use to describe their results. In particular the mentions of "breakthrough in gene regulatory network construction" are a bit over-the-top.

Response: We agree with the reviewer and thank them for this suggestion.

Changes: We have toned down the wording in the manuscript. For example, we have removed the word “breakdown” and removed “very” from “very precise”. Please also refer to the manuscript version with tracked changes.

- The methodology is relatively well described in terms of what is done, but is not so well described in terms of why it is done the way it is. The whole T_Z function is a bit mysterious for me. It seems to be some sort of matrix multiplication with normalization built-in. I do not understand the motivation for using it to generate both the R and A matrices or the reason that they are then averaged to give the delta to the new iteration of the W matrix. Also, the notion that the C and P networks are also updated is unclear. Without the understanding of what is the reason to this particular approach, the whole work seems arbitrary and even if the results are quite impressive, it is not really a step forward for the field.

Response: We value the reviewer’s feedback and have carefully considered their suggestion. We had included detailed information about how the message passing approach in PANDA works mathematically, as Reviewer 1 in the previous round of reviews had asked for more details related to the PANDA algorithm. PANDA uses a modified version of Tanimoto similarity that can be applied to compare network edge weights and calculate the similarity of the three input networks based on their targeting profiles. For an in-depth comprehension of the PANDA message passing algorithm’s methodology, we encourage the reviewer to refer to the complete description accessible at: <https://doi.org/10.1371/journal.pone.0064832.s005>.

As we implemented the original message passing framework that is described in the PANDA paper into *SCORPION*, we believe it is better not to include these details in the main text of our manuscript, and have thus moved this description to the Supplemental Material. In addition, to enhance the clarity of the PANDA algorithm, we have, we have incorporated additional detailed information on the algorithm in the Supplemental Material. Furthermore, we have emphasized the distinctions between *SCORPION*, PANDA and alternative methods for reconstructing gene regulatory networks from single-cell RNA-seq data.

Changes: We made changes to two sections of the manuscript. We added a supplementary section, titled: *Additional description of the SCORPION algorithm*, providing additional details about the PANDA message passing algorithm, and added more details to the discussion highlighting differences between the implementations of PANDA and *SCORPION*. For the changes made to the discussion, please refer to the version of the manuscript with track changes.

- The part regarding the *Hnf* factors changes upon DKO experiment are not really very convincing to me. Indeed the authors observe a shift downwards in the distribution of out-degrees of the genes knocked out. On one hand - there is a significant difference in the direction that we expect. On the other hand - these genes are knocked out in the actual data. Why do they have any out degrees in these data at all? (and BTW. I do not understand what are these distributions over. I would expect an outdegree, defined as a sum of all weight out of a single node to be a number, not a distribution).

Response: We appreciate the reviewer’s input, which helped identify a terminology error in our manuscript. Specifically, in the context of the section *SCORPION accurately detects changes in transcription factor activity and their impact on target genes* and Figure 2, when referring to out-degrees, we actually meant the distribution of edge weights connecting the transcription factor to all other genes. We apologize for the confusion that this has caused and have corrected this in the manuscript (please see below). Concerning the recovery of signals for perturbed transcription factors, we would like to clarify that the dataset provided by the authors still contains expression levels above the detection threshold for both transcription factors. This is because knock-outs are often not complete, and the transcription factor may be expressed to some degree in some cells. For instance, *Hnf4 α* expression is found in 16% of the cells in the *Hnf4 α γ ^{WT}* dataset and 18% in the *Hnf4 α γ ^{DKO}* dataset. Similarly, *Hnf4 γ* expression is present in 16% of the cells in the *Hnf4 α γ ^{WT}* dataset and 22% in the *Hnf4 α γ ^{DKO}* dataset. This pattern is, in fact, typical in experimental knockouts, where the aim is to render the protein non-functional by typically affecting a single transcript. In evaluating *SCORPION*’s predictions, we could actually use this to our advantage, as it allowed us to evaluate the correspondence of TF expression with their targeting across individual cells.

Changes: The text in the subsection *SCORPION accurately detects changes in transcription factor activity and their impact on target genes*, as well as Figure 2 and its corresponding legend, have been revised.

- The third part, concerned with the colorectal cancer data is quite interesting. The authors show that they are quite creative in their use of the results of the *SCORPION* method. And some of the results they show (like the differential network analysis or regulatory differences between left- and right- sided cancer cells are very interesting and quite a creative way to use the tool they have. However, as this is a methodological paper, I do not think that the method is described well enough. The use of the T_Z function and the whole process needs to be explained better, and - even more importantly - motivated for the paper to be a proper step forward in the field.

Response: We thank the reviewer for their positive evaluation of our application of *SCORPION* to colorectal cancer data. We appreciate the reviewer’s input regarding the description of the methodology, and have

thoroughly taken their suggestion into account. To improve the clarity of the PANDA algorithm, we have added more detailed information. Additionally, we highlighted the differences between *SCORPION*, PANDA, and other methods used for reconstructing gene regulatory networks from single-cell RNA-seq data. Please refer to question 2 by the same reviewer.

Changes: We made changes to two sections of the manuscript. The results section now provides additional details about the PANDA message passing algorithm, while the discussion highlights differences between the implementations of PANDA and *SCORPION*.

Decision Letter, second revision:

Dear Dr. Kuijjer,

Thank you for submitting your revised manuscript "Population-level comparisons of gene regulatory networks modeled on high-throughput single-cell transcriptomics data" (NATCOMPUTSCI-23-0230B). It has now been seen by the original referees and their comments are below. The reviewers find that the paper has improved in revision, and therefore we'll be happy in principle to publish it in Nature Computational Science, pending minor revisions to satisfy the referees' final requests and to comply with our editorial and formatting guidelines.

TRANSPARENT PEER REVIEW

Nature Computational Science offers a transparent peer review option for original research manuscripts. We encourage increased transparency in peer review by publishing the reviewer comments, author rebuttal letters and editorial decision letters if the authors agree. Such peer review material is made available as a supplementary peer review file. **Please remember to choose, using the manuscript system, whether or not you want to participate in transparent peer review.** Please note: we allow redactions to authors' rebuttal and reviewer comments in the interest of confidentiality. If you are concerned about the release of confidential data, please let us know specifically what information you would like to have removed. Please note that we cannot incorporate redactions for any other reasons. Reviewer names will be published in the peer review files if the reviewer signed the comments to authors, or if reviewers explicitly agree to release their name. For more information, please refer to our [FAQ page](https://www.nature.com/documents/nr-transparent-peer-review.pdf).

Thank you again for your interest in Nature Computational Science. Please do not hesitate to contact me if you have any questions.

Best,
Fernando

--

Fernando Chirigati, PhD
Chief Editor, Nature Computational Science
Nature Portfolio

ORCID

Reviewer #3 (Remarks to the Author):

The authors have done a great job addressing my comments. While some aspects are now clearer in the manuscript, the new randomization experiment also clearly strengthens the paper and shows that prior information is indeed contributing positively to the network inference task.

Reviewer #4 (Remarks to the Author):

The authors have addressed my concerns and I find the paper appropriate for publication in its current form.

In particular I appreciate the changes done in the Discussion section to better reflect the character of the paper's contribution. Also, clarifications in the text are a big improvement in the paper readability for the broad audience of Nature Computational Science.

Final Decision Letter:

Date: 17th January 24 11:58:07

Last Sent: 17th January 24 11:58:07

Triggered By: Fernando Chirigati

From: fernando.chirigati@us.nature.com

To: marieke.kuijjer@ncmm.uio.no

BCC: rjsproduction@springernature.com,fernando.chirigati@us.nature.com,computationalscience@nature.com

Subject: Decision on Nature Computational Science manuscript NATCOMPUTSCI-23-0230C

Message: Dear Dr Kuijjer,

We are pleased to inform you that your Article "Population-level comparisons of gene regulatory networks modeled on high-throughput single-cell transcriptomics data" has now been accepted for publication in Nature Computational Science.

Once your manuscript is typeset, you will receive an email with a link to choose the appropriate publishing options for your paper and our Author Services team will be in touch regarding any additional information that may be required.

Please note that *Nature Computational Science* is a Transformative Journal (TJ). Authors may publish their research with us through the traditional subscription access route or make their paper immediately open access through payment of an article-processing charge (APC). Authors will not be required to make a final decision about access to their article until it has been accepted. [Find out more about Transformative Journals](https://www.springernature.com/gp/open-research/transformative-journals)

Authors may need to take specific actions to achieve a

<https://www.springernature.com/gp/open-research/funding/policy-compliance-faqs> > **compliance** with funder and institutional open access mandates. If your research is supported by a funder that requires immediate open access (e.g. according to [Plan S principles](https://www.springernature.com/gp/open-research/plan-s-compliance)) then you should select the gold OA route, and we will direct you to the compliant route where possible. For authors selecting the subscription publication route, the journal's standard licensing terms will need to be accepted, including [self-archiving policies](https://www.springernature.com/gp/open-research/policies/journal-policies). Those licensing terms will supersede any other terms that the author or any third party may assert apply to any version of the manuscript.

Acceptance of your manuscript is conditional on all authors' agreement with our publication policies (see <https://www.nature.com/natcomputsci/for-authors>). In particular your manuscript must not be published elsewhere and there must be no announcement of the work to any media outlet until the publication date (the day on which it is uploaded onto our web site).

Before your manuscript is typeset, we will edit the text to ensure it is intelligible to our wide readership and conforms to house style. We look particularly carefully at the titles of all papers to ensure that they are relatively brief and understandable.

Once your manuscript is typeset, you will receive a link to your electronic proof via email with a request to make any corrections within 48 hours. If, when you receive your proof, you cannot meet this deadline, please inform us at rjsproduction@springernature.com immediately.

If you have queries at any point during the production process then please contact the production team at rjsproduction@springernature.com.

We welcome the submission of potential cover material (including a short caption of around 40 words) related to your manuscript; suggestions should be sent to Nature Computational Science as electronic files (the image should be 300 dpi at 210 x 297

mm in either TIFF or JPEG format). We also welcome suggestions for the Hero Image, which appears at the top of our [home page](http://www.nature.com/natcomputsci); these should be 72 dpi at 1400 x 400 pixels in JPEG format. Please note that such pictures should be selected more for their aesthetic appeal than for their scientific content, and that colour images work better than black and white or grayscale images. Please do not try to design a cover with the Nature Computational Science logo etc., and please do not submit composites of images related to your work. I am sure you will understand that we cannot make any promise as to whether any of your suggestions might be selected for the cover of the journal.

Best,
Fernando

--

Fernando Chirigati, PhD
Chief Editor, Nature Computational Science
Nature Portfolio

P.S. Click on the following link if you would like to recommend Nature Computational Science to your librarian: <https://www.springernature.com/gp/librarians/recommend-to-your-library>

** Visit the Springer Nature Editorial and Publishing website at <http://editorial-jobs.springernature.com> for more information about our career opportunities. If you have any questions please click [here](mailto:editorial.publishing.jobs@springernature.com).**